# Increased expression of peptides from non-coding genes in cancer proteomics datasets suggests potential tumor neoantigens

Rong Xiang[1,2], Leyao Ma[2,3], Mingyu Yang[2], Zetian Zheng[2], Xiaofang Chen[2], Fujian Jia[2], Fanfan Xie[2], Yiming Zhou[4], Fuqiang Li [2,5], Kui Wu [2,5] & Yafeng Zhu [4✉]

Neoantigen-based immunotherapy has yielded promising results in clinical trials. However, it is limited to tumor-specific mutations, and is often tailored to individual patients. Identifying suitable tumor-specific antigens is still a major challenge. Previous proteogenomics studies have identified peptides encoded by predicted non-coding sequences in human genome. To investigate whether tumors express specific peptides encoded by non-coding genes, we analyzed published proteomics data from five cancer types including 933 tumor samples and 275 matched normal samples and compared these to data from 31 different healthy human tissues. Our results reveal that many predicted non-coding genes such as *DGCR9* and *RHOXF1P3* encode peptides that are overexpressed in tumors compared to normal controls. Furthermore, from the non-coding genes-encoded peptides specifically detected in cancers, we predict a large number of "dark antigens" (neoantigens from non-coding genomic regions), which may provide an alternative source of neoantigens beyond standard tumor specific mutations.

[1] BGI Education Center, University of Chinese Academy of Sciences, Shenzhen, China. [2] BGI-Shenzhen, Shenzhen, China. [3] Southeast University, Nanjing, China. [4] Guangdong Provincial Key Laboratory of Malignant Tumor Epigenetics and Gene Regulation, Medical Research Center, Sun Yat-Sen Memorial Hospital, Sun Yat-Sen University, Guangzhou, China. [5] Guangdong Provincial Key Laboratory of Human Disease Genomics, Shenzhen Key Laboratory of Genomics, BGI-Shenzhen, Shenzhen, China. ✉email: zhuyaf@mail.sysu.edu.cn

Recently, many mass spectrometry-based proteomics studies have reported the identification of peptides from noncoding regions of the human genome[1–5]. Some peptides have been identified from genomic regions in close proximity to protein-coding genes, indicating incorrect annotation of exon boundaries or exons. Other noncoding gene-encoded peptides have been identified from currently annotated noncoding sequences, such as pseudogenes, long noncoding RNAs (lncRNAs), protein-coding gene untranslated regions, alternative reading frames, or the antisense strand.

It is conventionally believed that pseudogenes have lost their protein-coding functions due to accumulated deleterious mutations[6]. Recently, the analysis of RNA-seq data from cancer cell lines and tumors has shown active transcription of pseudogenes in different cell lineages and cancer types, and some pseudogenes have shown cancer-specific expression patterns when compared to normal tissues[7–9]. In addition to RNA level detection of pseudogene expression, several independent proteomics studies have identified peptide evidence of pseudogene and lncRNA translation in normal tissues and cancer cell lines[1–3]. However, it has not been systematically investigated if predicted noncoding genes encode peptides that can be found in tumor tissues or whether translation of noncoding genes is a sporadic event, or if it is specifically regulated in different types of tumors.

Here, we analyzed publicly available proteomics data from 5 cancer types, including 933 tumor samples and 275 matched normal samples, and 31 different healthy human tissues using a previously developed proteogenomics pipeline[4]. With these data, we aimed to identify which noncoding sequences, including pseudogenes and lncRNAs, are actively translated in healthy and tumor tissues and if they exert tissue-specific or cancer-specific expression. Secondly, a published study by Laumont et al.[10] detected more tumor-specific antigens from noncoding regions compared to mutations in protein-coding regions. They are in vivo mouse experiments demonstrated that immunization against these noncoding region peptides could prevent tumors in mice that had been transplanted with oncogenic cancer cells. Inspired by this, our second goal was to investigate whether these non-coding region-encoded peptides have any predicted affinity with MHC class I molecules as potential new cancer neoantigens.

## Results

### Construction of a proteogenomics search database.
We downloaded proteomics data collected from 40 normal samples from 31 healthy tissues, 933 tumor samples, and 275 tumor-adjacent normal samples from the PRoteomics IDEntifcations (PRIDE) database and National Cancer Institute Clinical Proteomic Tumor Analysis Consortium (CPTAC) Data Portal[11,12]. The five types of cancer investigated were breast cancer (BRCA), clear cell renal cell carcinoma (CCRCC), colon cancer (COAD), and ovarian cancer, and uterine corpus endometrial carcinoma (UCEC). The number of samples in each dataset is presented in Fig. 1a. The detailed annotations of the downloaded datasets and sample information are included in Supplementary Data 1 (Table 1).

To search the proteomics data, we first constructed a core database, including ENSEMBL human proteins, CanProVar 2.0 variant peptides, and peptide sequences from three frame translations of annotated pseudogenes from GENCODE v28 and lncRNA from LNCpedia 4.1[13–16]. This core database was used as a search database for the proteomics data of healthy tissues. As for different cancer datasets, a collection of cancer mutations was downloaded from the CGDS[17]. These mutations were then converted to mutant protein sequences using customized scripts and supplemented to the search database of the corresponding cancer type (see details in "Methods"). The

proteogenomics search was performed using an updated version based on our previously published pipeline[4] (Supplementary Fig. S1).

### Majority of novel peptides identified from pseudogenes are homologous to house-keeping genes.
In total, we identified 7882 and 9013 novel peptides from 31 normal tissues and five cancer types at a 1% class-specific false-discovery rate (FDR), respectively. Novel peptides/coding loci were defined as peptide/genomic sequences that are absent in annotated protein/coding gene databases (Uniprot human reference proteome plus GENCODE v28 human protein database)[18]. We summarized the number of unique peptides per novel coding locus for 31 healthy tissues, and the CPTAC datasets (including both 933 tumor samples and 275 tumor-adjacent normal samples), respectively (Fig. 1b, c). We then divided the novel loci into three groups according to the number of unique peptides by which they were supported. After removing loci supported by only a single peptide, in total 220 and 687 novel coding loci (corresponding to 603 and 2320 unique peptides) were identified in the healthy tissue data and CPTAC datasets (Fig. 1d), respectively (detailed annotations of novel coding loci are provided in Supplementary Data 1 (Table 2 and Table 3)).

Next, we annotated the identified novel peptides detected from the healthy tissues dataset and CPTAC datasets based on their origin, including pseudogenes; lncRNAs; untranslated regions, introns, and exons of protein-coding genes (alternative reading frame); upstream and downstream regions (1 kb distance to closest UTR) of protein-coding genes; spanning intron–exon junctions of protein-coding genes; and retroelements (Fig. 1e). In a paired t-test comparison, the CPTAC datasets and the healthy tissues showed no significant difference in the percentage of novel coding loci detected in different genomic regions. Further, consistent with the findings in Kim et al.[19] and our previous work[4], the majority of novel peptides were from translating pseudogenes. LncRNAs were the second major source of identified novel peptides. The low percentage of novel peptides detected from lncRNAs is in line with a previous study by Guttman et al.[20], in which a comprehensive analysis of ribosomal profiling data provided supporting evidence that the large majority of lncRNAs do not encode proteins.

Because pseudogenes have high sequence similarity to their parental genes, we annotated the translated pseudogenes based on the functions of their parental genes (Fig. 1f, Supplementary Fig. S2a). Consistent with the findings revealed by a previous RNA-seq data analysis[7], the frequently detected pseudogenes in healthy tissues and CPTAC datasets were homologous to house-keeping genes such as cytoskeleton proteins (actin, keratin, and tubulin), ribosomal proteins, nuclear ribonucleoproteins, heat shock proteins, and eukaryotic translation elongation factor, peptidylprolyl isomerase (Fig. 1f, Supplementary Data 1, Table 4). These pseudogene peptides included 428 and 1970 novel peptides, comprising 70.9% and 84.9% total novel peptides, detected from the healthy tissues and CPTAC datasets, respectively.

### Proteomics detects ubiquitous and tissue-specific translation of pseudogenes and lncRNAs.
Previous proteogenomics studies have mainly characterized protein level alterations from genomics aberrations including copy number variations and missense mutations[21–23]. Our recent work investigated the tissue-specific expression of noncoding gene-encoded peptides in five different human tissues[4]. Here, we extended this analysis in a comprehensive proteomics dataset of 31 different tissues[5]. We quantified the identified novel peptides by extracting MS1 maximum peak intensity using moFF[24]. We limited the analysis to novel coding

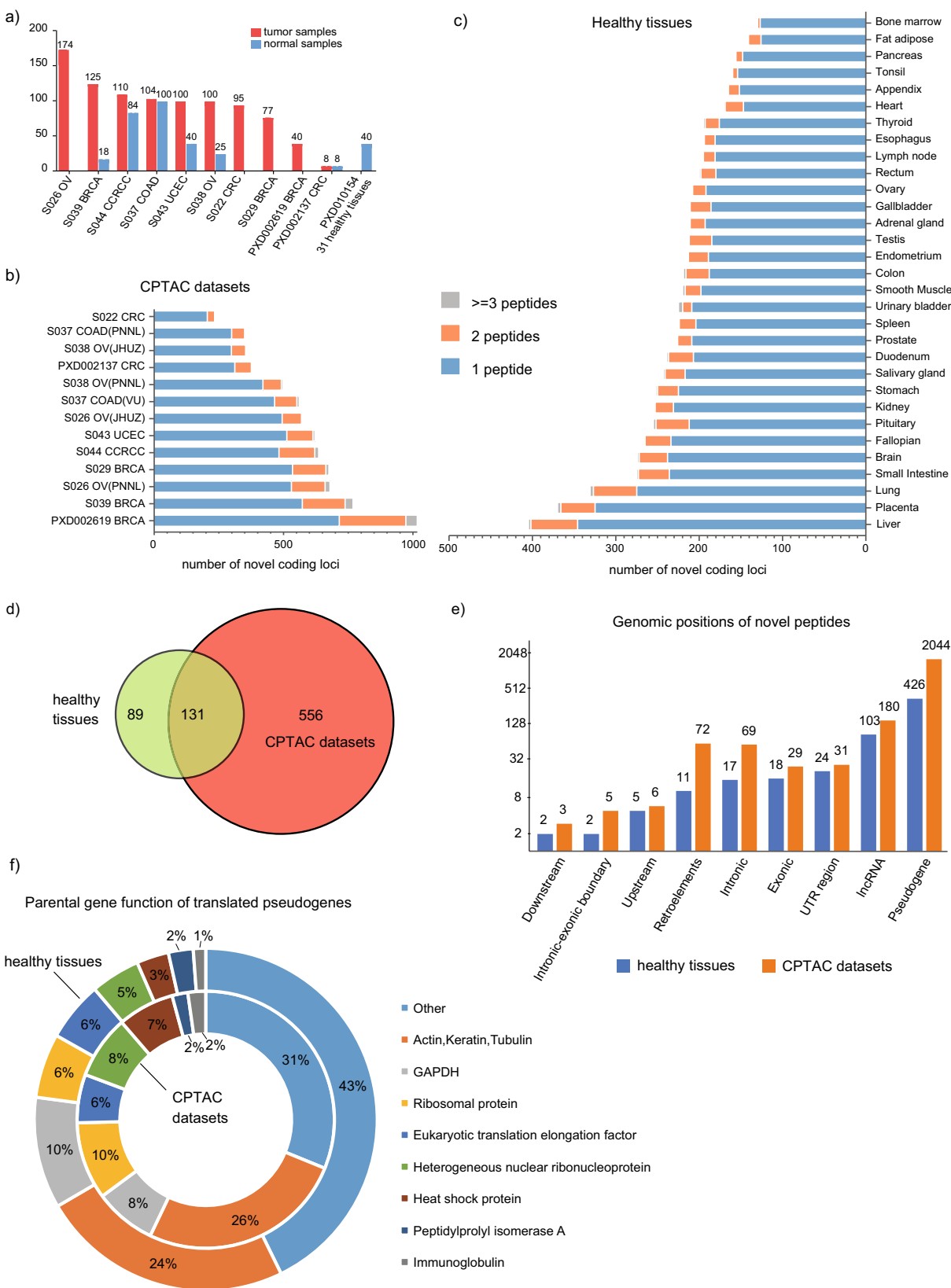

loci with at least two unique peptides. Our analysis identified three groups of novel coding loci expression: 12 ubiquitous (expressed in at least 15 tissues), 93 nonspecific (expressed in 2–15 tissues, robustly translated in one or two tissues but frequently translated at lower levels in other tissues), and 114 with

tissue-specific expression (Fig. 2). The pseudogene expression profile we observed was different from the RNA-seq study, where the majority of expressed pseudogenes were identified as nonspecific[7]. We speculated that many non-specific and lowly expressed pseudogenes were stochastically detected in tissues with

**Fig. 1 Noncoding gene-encoded peptides detected from normal tissues and CPTAC datasets. a** Type and number of samples used in this study. **b** The number of novel coding loci detected in CPTAC datasets. **c** The number of novel coding loci detected in 31 healthy tissues (peptides are grouped to one locus if they are encoded by the same noncoding gene). **d** Venn diagram shows the overlap of novel coding loci (≥2 unique peptides detected) between healthy tissues and CPTAC datasets. **e** Annotation of genomic positions where noncoding gene-encoded peptides are detected. Pseudogene: all categories of pseudogene (if novel peptide belongs to pseudogene, we will not count it again in other categories). lncRNA: ncRNA. Exonic: coding gene's exon, not at the canonical reading frame. Intronic: coding genes intron. Intronic–exonic boundary: peptide spanning over coding gene's exon–intron boundary. UTR region: untranslated region of the coding gene. Upstream: upstream of the coding gene. Downstream: downstream of the coding gene. **f** Annotation of parental genes' function of translated pseudogenes.

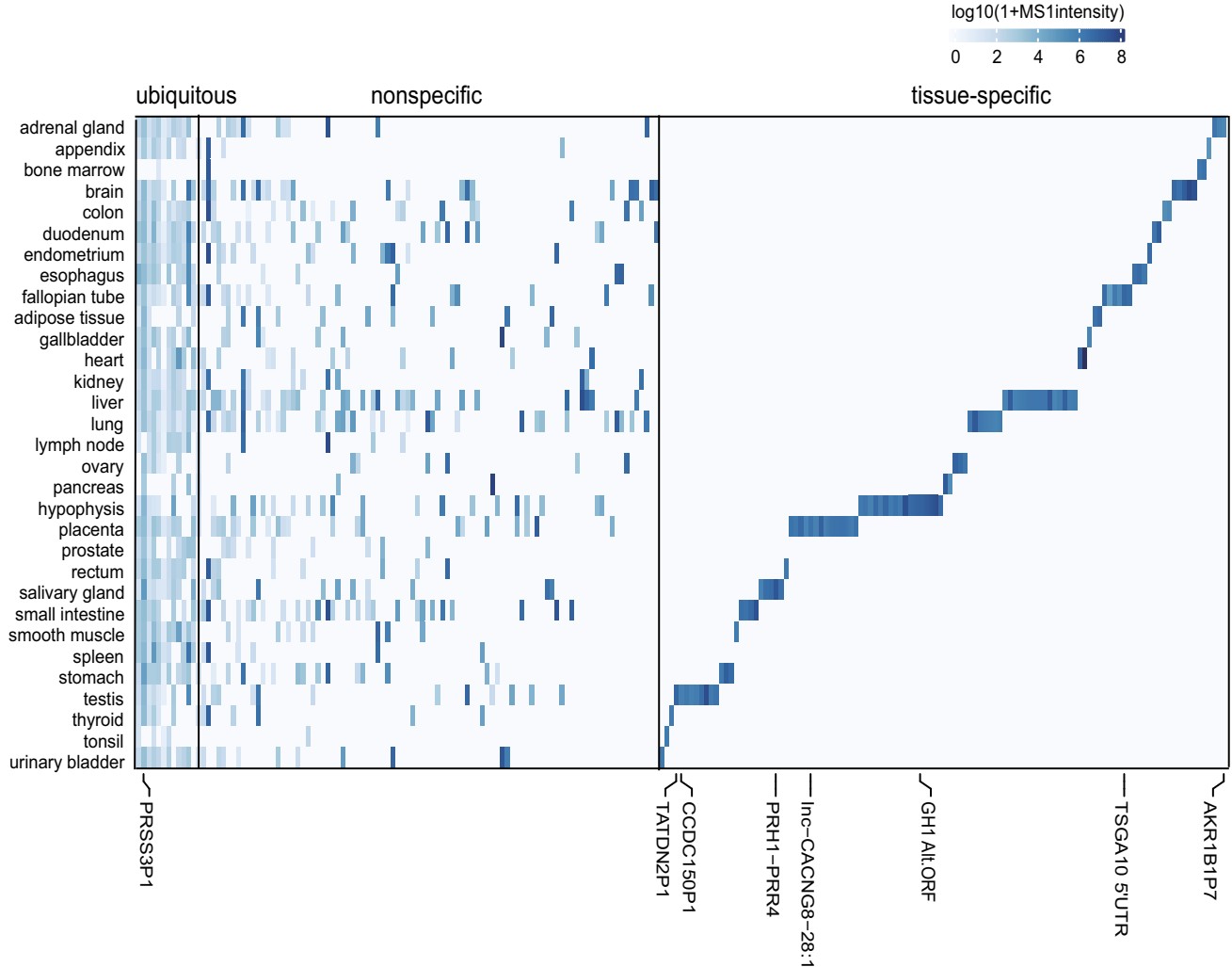

**Fig. 2 Heatmap of MS1 intensity of novel coding loci encoded peptides detected in normal tissues.** Heatmap of novel coding loci expression sorted based on tissue-specific expression shows ubiquitously (left), nonspecific (middle), and tissue-specific (right) expressed novel coding loci.

only one sample analyzed (24 of 31 tissues have only one sample), consequently increasing the number of tissue-specific pseudogenes here (see Supplementary Data 1, Table 2, with representative tissue-specific pseudogenes/lncRNAs highlighted).

From the 31 healthy tissues dataset, we detected two previously reported tissue-specific non-coding gene translation products: testis-specific *TATDN2P1* (TatD DNase domain-containing 2 pseudogene 1, supported by two unique peptides) and placenta-specific lncRNA *lnc-CACNG8-28:1* (supported by eight unique peptides)[4]. In addition, several new tissue-specific non-coding genes were discovered (see Supplementary Data 1, Table 2). For example, ten unique peptides encoded by a lncRNA, *lnc-AFF3-13:1*, located in the 5′ UTR of gene *TSGA10* were detected in fallopian tissue. Six unique peptides from a *PRH1-PRR4* read-

through transcript were detected in the salivary gland. Pseudogene *CCDC150P1* was detected with five unique peptides in testis, and this pseudogene *CCDC150P1* transcript is also specifically expressed in testis according to GTex data (Supplementary Fig. S2b). Interestingly, in both pituitary tissue samples, peptides were identified from a lncRNA that overlaps with the coding region of a pituitary specific protein-coding gene, *GH1*, but in a noncanonical reading frame (see annotated spectra in Supplementary Data 2). Our data indicate that *GH1* may have dual coding frames that encode unknown new proteins.

We compared our proteomics results with two recent studies that used ribosomal profiling and full-length mRNA sequencing to search translated noncoding genes in multiple cell types and cancer cell lines[25,26]. Lu et al. identified 2969 translating non-

coding genes from mRNA sequencing and ribosomal profiling data, and mass spectrometry detected 10% (308) noncoding gene-encoded new proteins (372 unique peptides). Among these new proteins, 59 were also identified in our results (See Supplementary Data 1, Table 5). These include *MCTS2P, MKKS* 5′ UTR ORF, *LINE-1* ORF1, and *PA2G4P4*. In comparison, only eight novel CDS were found in common between Chen et al.[27] and our current study. This could be due to the sample difference since their novel CDS were identified from induced pluripotent stem cells (iPSCs), iPSC-derived cardiomyocytes, and human foreskin fibroblasts. Of note, these common novel CDS include *MCTS2P*, *STARD10* 5′ UTR ORF, and *TSGA10* 5′ UTR ORF.

**Overlap of detected non-coding gene translation in different samples and datasets**. We analyzed the overlap of detected novel coding loci in different samples within each study (Fig. 3a). We divided the novel coding loci into four groups by the percentage of samples in which they were identified. For example, the dataset PXD002619 produced the largest number of novel coding loci, but two-thirds were identified in fewer than 25% of samples. On average, one-third of all novel loci were identified in more than 50% of samples.

Among different CPTAC datasets (in total 13 datasets covering five cancer types), 46% of pseudogene identifications were repeatedly detected in at least two datasets. In comparison, only 16% of non-pseudogenes were detected in more than one dataset. Further analysis showed that 93% of pseudogenes that were identified commonly in 8–13 different datasets belong to housekeeping genes, which suggests pseudogenes derived from house-keeping genes are also ubiquitously expressed in different cancer types (Fig. 3b).

Apart from the ubiquitously expressed pseudogenes, many pseudogenes were recurrently detected in specific cancers. The notable examples were *RHOXF1P3* (Rhox homeobox family member 1 pseudogene 3) and *MCTS2P* (malignant T cell amplified sequence 2 pseudogenes) which were repeatedly detected from independent datasets of breast and ovarian cancers (Fig. 3c). The parental gene of *RHOXF1P3*, *RHOXF1*, is thought to inhibit cell apoptosis by activation of BCL-2[28]. *MCTS2* is an imprinted gene and only paternally expressed retrogene copy[29].

In addition to pseudogenes, we also found several long noncoding RNA-encoded peptides that were detected in specific cancers. For example, lncRNA *lnc-SERPIND1-41:10* were detected with ten unique peptides from different samples in CCRCC (Fig. 4c, Supplementary Data 1, Table 3). This lncRNA is located in the last intron of the noncoding RNA gene *DGCR9* (DiGeorge Syndrome Critical Region Gene 9, located on chromosome 22q11, see Supplementary Fig. S3). Our results present the first evidence to our knowledge that a potential novel coding locus in *DGCR9*'s last intron may encode a protein product in CCRCC.

Since pseudogene expression has been extensively analyzed at the transcript level using RNA-seq data[7,8] and the major biological functions of pseudogenes have been revealed at the RNA level, we wondered whether any pseudogenes expressed at the RNA level are translated into proteins. Therefore, we compared pseudogenes detected in our proteomics analysis with two major studies in which the expression of pseudogenes was investigated through RNA-seq analysis[7,8]. We found that the pseudogenes commonly detected in RNA and protein level are pseudogenes of house-keeping genes such as ribosomal proteins, GAPDH, cytokeratin, eukaryotic translation initiation factors, and heterogeneous nuclear ribonucleoprotein. In addition, pseudogenes corresponding to cancer-associated genes *HMGB1*, *VDAC1*, and *PTMA* reported in a previous RNA-seq study[7] were detected both in the healthy tissues and cancers in our proteomics

analysis (Supplementary Data 1, Tables 2 and 3). In comparison, many of the known functional pseudogenes such as *PTENP1* were not detected in these proteomics data. This was not unexpected since they are functional as ceRNA molecules regulating the expression of their parental genes[30]. Another example is the BRCA pseudogene *ATP8A2P1*, which showed high expression at the RNA level[7,8] but was not detected at the protein level in any of the BRCA proteomics data, suggesting this pseudogene may only exert functions at the RNA level.

**Differential expressed noncoding gene-encoded peptides between tumor and normal tissue**. We investigated if certain pseudogene/lncRNA-encoded peptides had elevated expression in tumors in the colorectal cancer (CRC) dataset with 8 paired CRC samples and matched normal tissues (PXD002137)[27]. In this dataset, 73 pseudogenes and lncRNAs identified were supported by multiple peptides. Unsupervised clustering of these 73 pseudogenes and lncRNAs by the centered $\log_2$ intensity is shown in Fig. 4a. A paired *t* test analysis found 11 of the pseudogenes/lncRNAs were significantly upregulated in tumors compared to matched normal tissues. For example, *lnc-KMT5B-20:1*, *lnc-NANOGP8-26:6*, and *RP11-351N4.2* are upregulated in CRC compared to matched normal tissues (Fig. 4b).

In other cancer datasets, we also detected several noncoding gene-encoded peptides with increased expression in tumors. For example, the peptides encoded by lncRNA *lnc-SERPIND1-41:10* (*DGCR9* intron) showed significantly higher expression levels in CCRCC compared to adjacent normal tissues (Fig. 4c). In UCEC, peptides detected from the 5′ UTR or noncanonical reading frame of the protein-coding genes *TSGA10*, *NPLOC4*, *MKKS*, and *MUC1* were more abundant in tumors compared to normal tissues (Fig. 4d). Similarly, increased expression of peptides from *MKKS* 5′ UTR was also detected in another CRC dataset (Fig. 4e).

In the two CPTAC BRCA datasets, the pseudogene *RHOXF1P3* was identified with eight and seven unique peptides, respectively, covering 89% of amino acid sequences of the open reading frame encoded by this pseudogene (Fig. 5a). More interestingly, the peptides encoded by pseudogene *RHOXF1P3* were upregulated (2- to 16-fold) in a subset of BRCA patients both in the CPTAC BRCA Discovery and Confirmatory cohorts (Fig. 5b, c)[21]. In addition, *RHOXF1P3*-encoded peptides were also detected in two ovarian cancer patients (Fig. 5d). We then analyzed the expression of *RHOXF1P3* in a published RNA-seq dataset including 63 breast tumors and 10 adjacent normal tissues, which also showed upregulated expression of *RHOXF1P3* in tumor samples (Fig. 5e). Together, our results demonstrated that pseudogene *RHOXF1P3* is not only translated, but also upregulated in a subset of breast tumors.

Finally, we detected peptides from the 5′ UTR of *STARD10*, which also displayed higher abundance in a subset of breast tumors (Fig. 5f). *STARD10* is a lipid transfer protein and this protein has been previously reported to be overexpressed in BRCAs and correlate with ErbB2/Her2 status[31]. Our data suggest that this gene may use an upstream non-AUG start codon to initiate translation in a subset of breast tumors.

**LINE-1 retrotransposon ORF1 encoded peptides show higher expression in tumors**. As evidenced in many studies, cellular mechanisms that repress the expression of repetitive DNA are disrupted in cancer cells. Overexpression of satellite repeats was previously observed in pancreatic and other epithelial cancers[32,33]. This phenomenon correlates with the overexpression of the long interspersed nuclear element 1 (LINE-1) retrotransposon, which is suggested as a hallmark of many cancers[33].

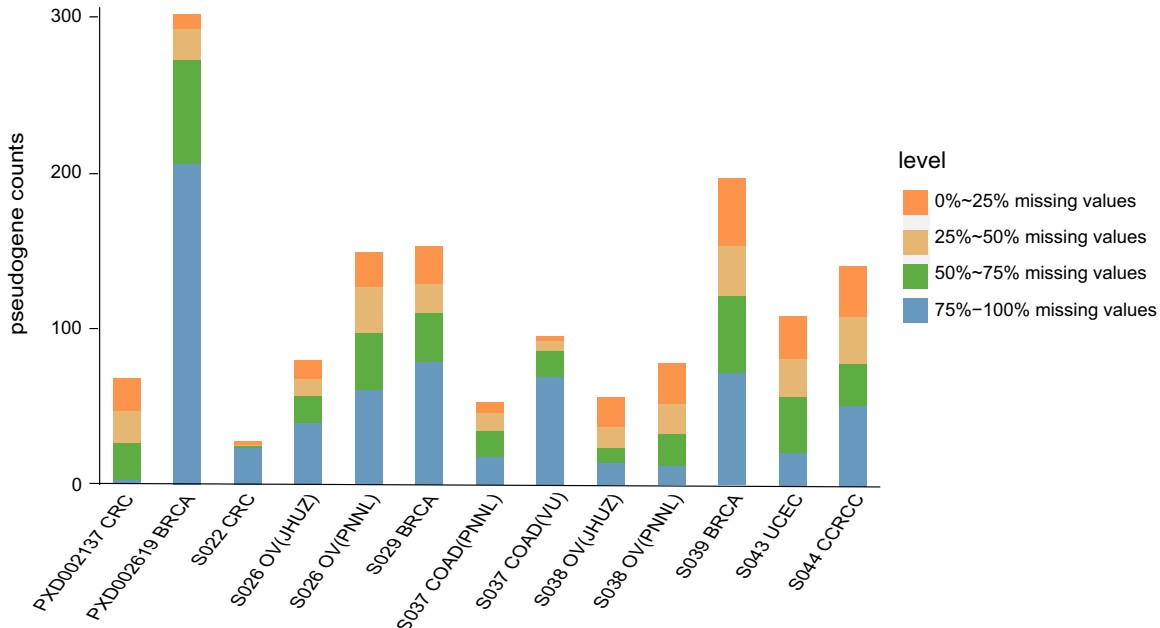

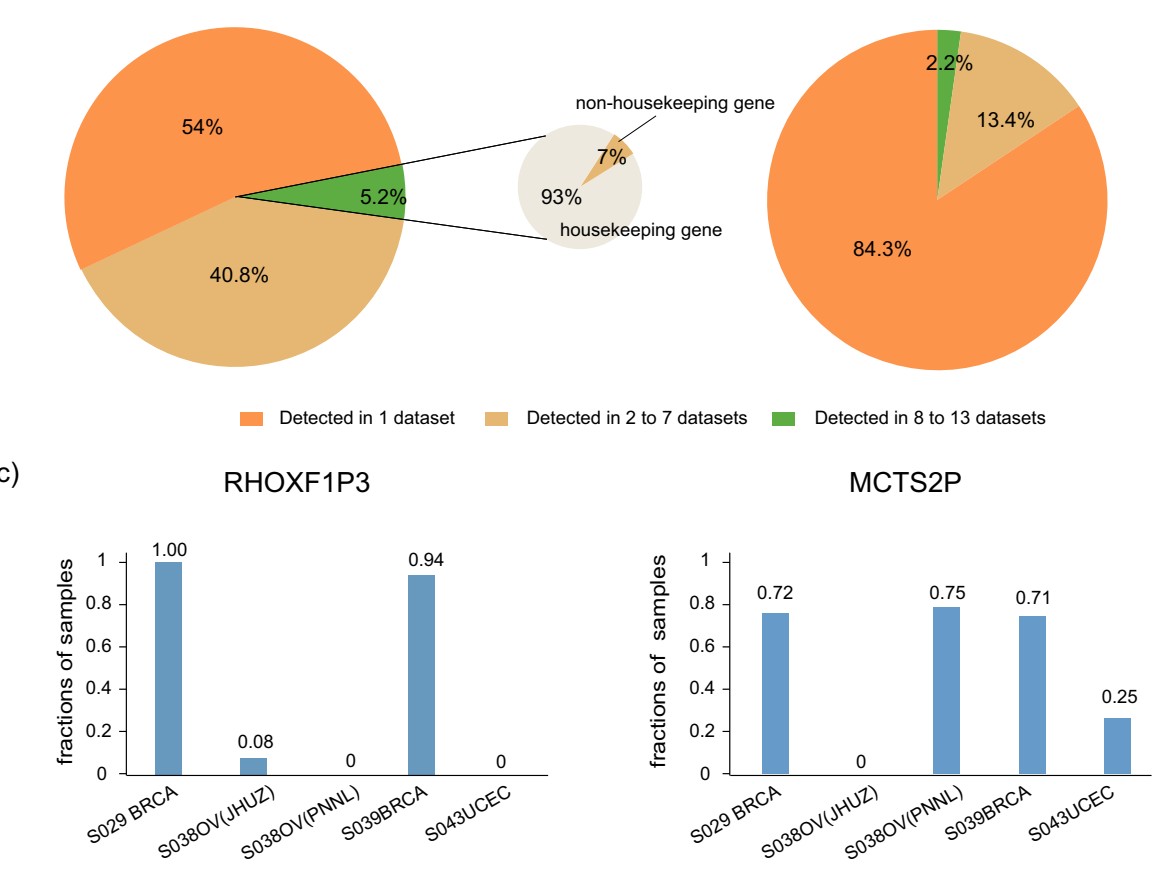

**Fig. 3 Overlap of detected noncoding gene translation in different samples and datasets. a** The overlap of detected novel coding loci in different samples. For example, red bar indicates the number of novel coding loci that are only missing in 0–25% of the samples, while the blue bar indicates the number of novel coding loci missing in 75–100% of the samples. For iTRAQ or TMT labeled data, a valid value or a missing value is used to determine if the locus is detected in the corresponding sample or not. **b** Overlap of pseudogenes and non-pseudogenes between CPTAC datasets. **c** The percentage of samples detected with *RHOXF1P3* and *MCTS2P*.

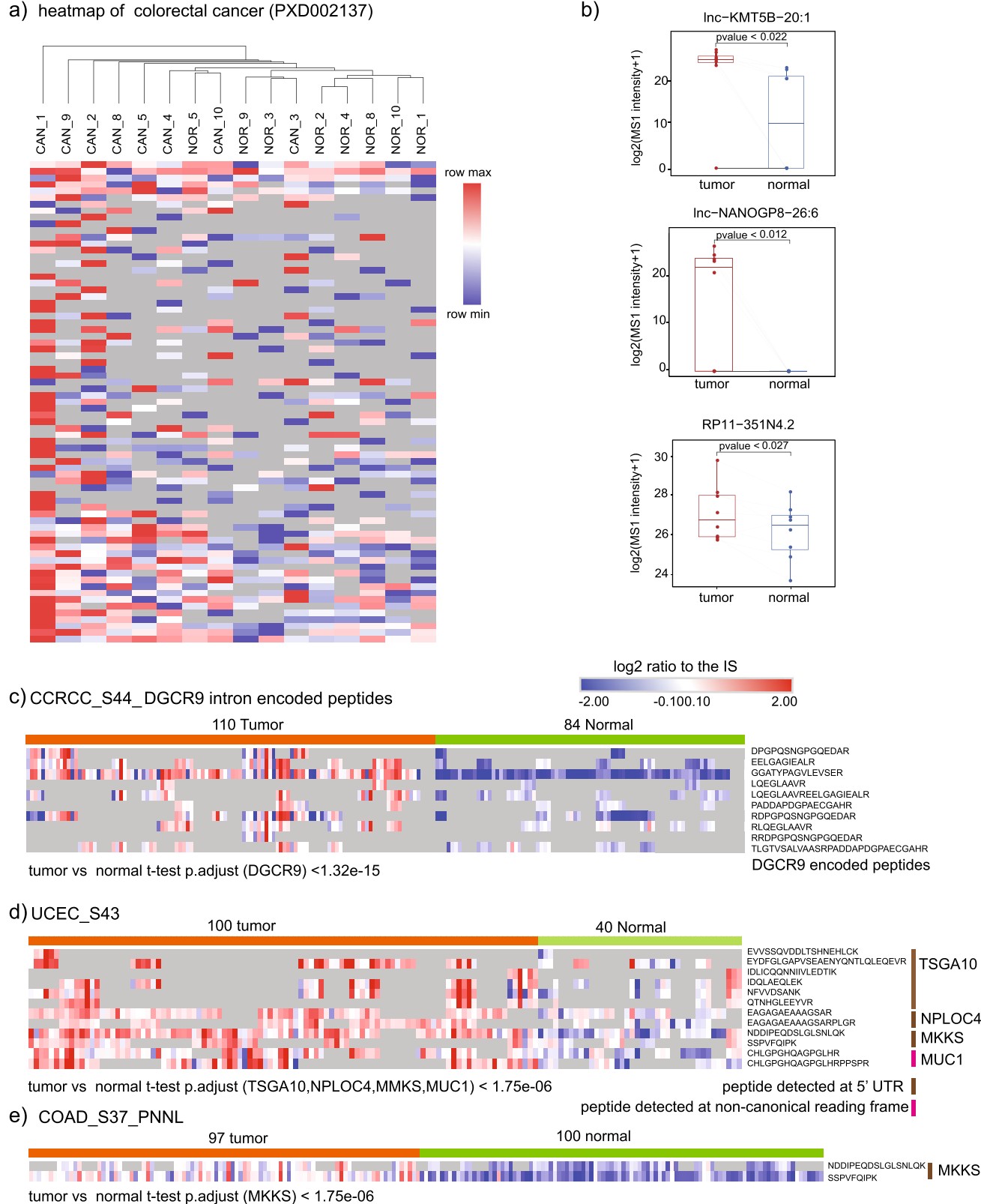

**Fig. 4 Differential expressed noncoding gene-encoded peptides between tumor and normal. a** Heatmap of colorectal cancer (PXD002137). Heatmap was scaled by row value. **b** Boxplot of noncoding genes significantly differentially expressed between tumor and normal (paired *t* test, *p*.value < 0.05). **c** Relative expression of the ten unique peptides detected from *DGCR9* in tumor (CCRCC) and normal (*t* test *p*.adjust < 0.01). **d** Relative expression of peptides detected at 5′ UTR in tumor (UCEC) and normal. **e** Relative expression of peptides detected from *MKKS* 5′ UTR in tumor (COAD) and normal. CCRCC clear cell renal cell carcinoma, UCEC uterine corpus endometrial carcinoma, COAD colon cancer.

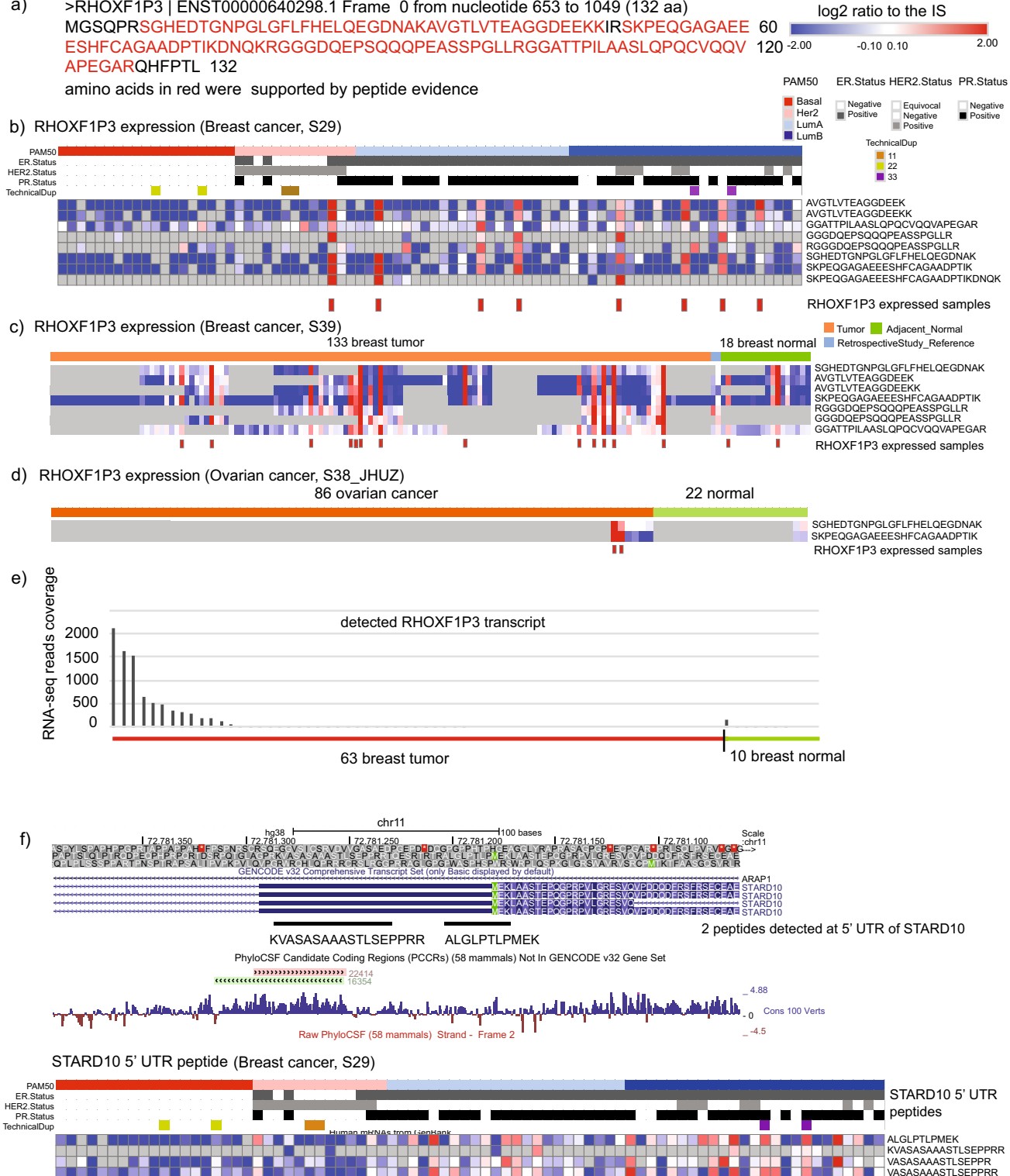

**Fig. 5 Differential expressed noncoding gene-encoded peptides in breast cancer. a** Predicted protein sequence of pseudogene *RHOXF1P3* (amino acids in red were supported by detected peptides). **b** Relative expression of *RHOXF1P3* encoded peptides in 77 breast tumors. Gray color boxes indicate missing values. **c** Relative expression of *RHOXF1P3* encoded peptides in 133 breast tumors and 18 adjacent normal tissues. **d** Relative expression of *RHOXF1P3* encoded peptides in 86 ovarian tumors and 22 normal ovarian tissues. **e** RNA seq read count of *RHOXF1P3* transcript in breast tumor and normal. **f** Relative expression of peptides detected at 5′ UTR of *STARD10* in 77 breast tumors.

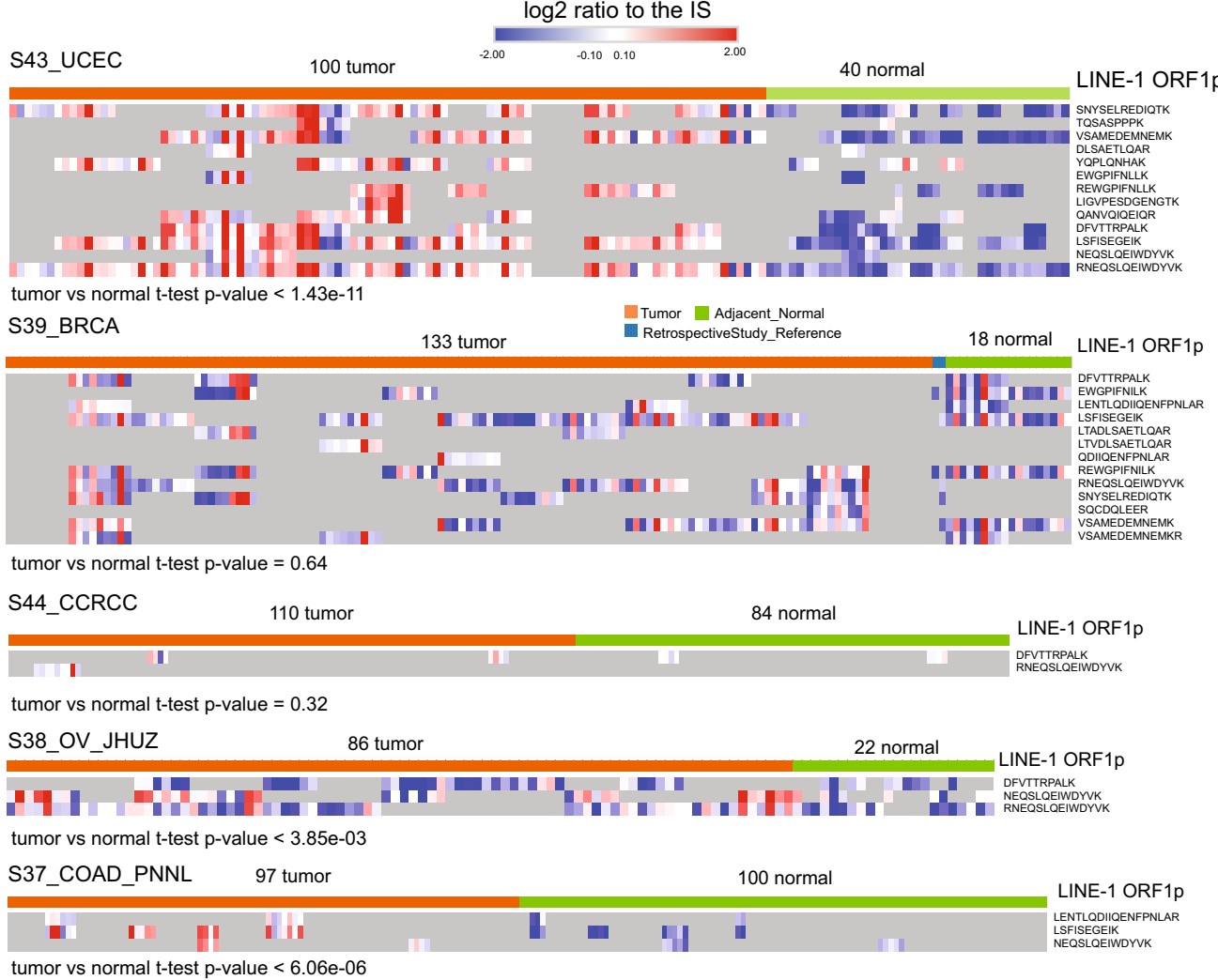

**Fig. 6 Relative abundance of LINE-1 ORF1 encoded peptides in tumor and normal.** The heatmap shows the log2 relative abundance of LINE-1 ORF1 encoded peptides in five different cancers. Gray boxes indicate missing values. BRCA breast cancer, OV ovarian cancer, CCRCC clear cell renal cell carcinoma, UCEC uterine corpus endometrial carcinoma, COAD colon cancer.

In previous proteogenomics studies, peptides mapped to multiple genomic locations were often neglected. In our analysis, LINE-1 retrotransposon ORF1-encoded peptides were detected in different cancer datasets. LINE-1 RNA contains two non-overlapping open reading frames, encoding two proteins ORF1p and ORF2p. The expression level of ORF1p is 1000-10,000 times higher than ORF2p[34]. In the analyzed proteomics datasets, we detected ORF1p peptides from all five cancer types (Supplementary Data 1 Table 3). The quantitative analysis showed higher expression of the LINE-1 ORF1p encoded peptides in UCEC, ovarian cancer and COAD compared to their respective normal samples (Fig. 6). In comparison, LINE-1 ORF2p was not detected in our analysis. It corroborates findings from an antibody-based study which concluded that LINE-1 ORF2p expression is hardly detectable in human cancers[35]. Surprisingly, we also detected peptides of LINE-1 ORF1 in healthy tissues, including lung, ovary, and prostate (Supplementary Data 1, Table 2). Another independent study using RNA-sequencing data also observed widespread expression of retroelements in human somatic tissues[36]. This may be explained by a recent finding that LINE-1 activity becomes derepressed in senescent cells and healthy tissues could have senescent cells at old age[37].

**Noncoding region encoded peptides as a new class of cancer neoantigens.** Laumont et al.[10] demonstrated noncoding region encoded peptides can be used as a cancer vaccine to prevent tumor progression. Here, we try to predict if any of noncoding region-encoded peptides can be used as potential cancer neoantigens. T cells recognize and bind to a peptide–MHC complex in a complex process with many crucial steps. First, the abnormal proteins are hydrolyzed by proteases into peptide fragments in the cytoplasm, and then peptide fragments transported by the transporter associated with antigen processing (TAP) protein into the endoplasmic reticulum, where the peptide bind to an MHC molecule[38]. The NetCTLpan server integrates predictions of proteasomal C terminal cleavage, TAP transport efficiency, and MHC class I binding affinities, which take into account antigen processing and presentation[39]. Therefore, we used the NetCTL-pan server to predict the neoantigens.

We selected neoantigen candidates based on the following criteria: (1) The average expression of novel loci in tumor tissue was upregulated by 1.5 times compared with matched normal tissue (restricted to the datasets in which matched normal tissues are available); (2) peptides were supported by NetCTLpan predictions. By NetCTLpan prediction, 64 pseudogenes or

lncRNAs had at least one 9-mer peptide with predicted affinity ranked at threshold ≤0.5% (Supplementary Data 1, Table 6). These results suggest that there are a large number of candidate neoantigens in the noncoding regions. Of note, the 9-mer peptide (HEDTGNPGL) encoded by pseudogene *RHOXF1P3,* and the peptide (RLQEGLAAV) encoded by lncRNA *lnc-SERPIND1-41:10* (*DGCR9* intron) were predicted as neoantigens.

## Discussion

Protein-coding genes have been predominantly annotated based on RNA-level data[40]. Several studies integrating the ribosome profiling data to unbiasedly search all potential coding sequences led to the findings that many of the predicted non-coding regions of the human genome are translated[25,26,41–44]. Since then, non-coding region-encoded peptides have been detected in several large-scale mass spectrometry-based proteomics studies[4,45]. Here, we analyzed large datasets of high-quality mass spectrometry proteomics data from 31 healthy tissues and five cancer types. In summary, we detected peptide evidence of 220 and 687 novel coding loci in the healthy tissues and CPTAC datasets, respectively. To avoid potential false positives, we only analyzed novel coding loci supported by at least two peptides, resulting in a majority of the identifications being removed (Fig. 1). The inter-patient overlap of identified novel loci within each study showed approximately one-third of them were detected in over 50% of samples (Fig. 3). Since false positives could be still present, we thus annotated MS2 spectra of 87 peptides from major findings described in the manuscript (Supplementary Data 2). The annotated spectra revealed several suspicious identifications, but most of them have well-matched fragment ions.

According to our analysis, the majority of novel peptides were identified from pseudogenes, which can be categorized into eight major functional classes of house-keeping genes based on parental genes (Fig. 1). The quantitative analysis of 31 healthy tissue proteomics data revealed ubiquitous, nonspecific, and tissue-specific expression of translated non-coding genes (Fig. 2). We acknowledge the limitation that it is not a complete representation of all tissue types and most of the tissue types have only one sample. Thus, some tissue-specific pseudogenes detected here may arise stochastically.

The role of pseudogenes in cancer development and progression has been increasingly investigated, with some of them proposed as diagnostic and prognostic markers[46]. In our analysis, several peptides encoded by non-coding gene *lnc-SERPIND1-41:10* (*DGCR9* intron) were detected in CCRCC and showed higher expression compared to normal tissues. In addition, several pseudogenes have been identified repeatedly in specific cancers and supported by multiple unique peptides, such as *RHOXF1P3* and *MCTS2P*. Our results provide evidence that pseudogene *RHOXF1P3* is not only translated (identified with eight unique peptides) but also showed increased expression in a subset of breast tumors compared to normal tissues (Fig. 5). Whether these non-coding gene-encoded proteins/peptides are crucial for cancer cell survival requires further studies to investigate their biological roles. During the revision of this paper, a study using CRISPR-based screening by Prensner et al.[47] revealed that hundreds of functional proteins encoded by noncanonical open reading frames are essential for cancer cell survival. To enable others to explore our data for further research, we have shared all identified noncoding genes encoded peptides in the healthy tissues and CPTAC datasets.

Mutation-derived tumor-specific neoantigens could be recognized by infiltrated T cells to trigger their cytotoxic activity against tumor cells. Neoantigen-based immunotherapy has been tested and shown promising results in clinical trials[48]. However, missense mutation-derived neoantigens may have limited immunogenicity as their non-self feature relies on only one amino acid difference. In comparison, neoantigens derived from non-coding gene-encoded peptides termed as "dark antigens", that possess larger sequence differences to hold greater promise due to their high potential immunogenicity[49]. According to our analysis by the NetCTLpan, there are a large number of potential dark antigens predicted from the noncoding gene-encoded peptides. In the aspect of clinical applications, these potential dark antigens from noncoding regions, especially the ones specific to tumors, may provide a new class of tumor neoantigens possessing more potent immunogenic activity to be explored as cancer vaccine[10].

## Methods

**Data sets**. Liquid chromatography–mass spectrometry/mass spectrometry (MS) raw files of 40 normal samples from 31 healthy tissues, 933 cancer samples of five cancer types, and 275 tumor-adjacent normal samples were obtained from the PRIDE database and CPTAC Data Portal[11,12]. See Supplementary Data 1 and Table 1 for details.

**Database construction**. To search the proteomics data of the healthy tissues, we used a core database which contains the human protein database of Ensembl 92, CanProVar 2.0 variant peptides, and peptide sequences from the three-frame translation of annotated pseudogenes and lncRNAs[13,14]. Pseudogenes were downloaded from GENCODE v28 including both annotated and predicted[15]. LncRNAs were downloaded from LNCpedia 4.1[16]. For each cancer type, a collection of cancer mutations detected from previous whole genome sequencing and whole exome sequencing studies were downloaded from the Cancer Genomics Data Server (CGDS, http://www.cbioportal.org/datasets) and then converted to mutant peptide sequences using customized scripts and supplemented to the search database of corresponding cancer type[17,50].

**Identification and quantification of novel peptides by IPAW**. The proteogenomics search was performed using a previously published pipeline[4]. We made the following adjustments. (1) database indexing. Previously, database indexing is performed separately for each spectra input file. Database indexing is done only once in the new workflow, and all parallel processes share the same indexed database at the search step. It reduces many intermediate indexing files and searching time. (2) returning hg38 genomic coordinates. Previously, all peptides were mapped to hg19 genome assembly, now peptides are mapped to hg38 assembly. The pipeline and description is shared at Github (https://github.com/yafeng/pan-cancer-proteogenomics-analysis). Briefly, all MS/MS spectra were searched by MSGFPlus[51] in the target and decoy combined database. The decoy peptide was produced by reversing protein sequences in the target database. Target and decoy matched to known tryptic peptides (from Ensembl92 human proteins) were discarded before FDR estimation of novel peptides. Peptide matches to known proteins were removed through BLASTP analysis using a more complete known protein database (ENCODE28 + Ensembl92 + Uniprot.2018April)[52]. Peptides matched to mutant peptide sequences from non-synonymous SNPs or cancer mutations were removed and not treated as novel peptides. Then peptides mapping to multiple genomic loci were annotated using BLAT[53]. The retained novel peptides were considered for further analysis. Peptides detected from 31 healthy tissues were quantified by moFF tool[24], which extracts apex MS1 intensity from raw spectral files. Annotated spectra of peptides of interest (in total 123 spectra of 87 unique peptides) were provided as supplemental material (Supplementary Data 2). One can search specific peptide sequences and genes of interest in the PDF to inspect their MS2 spectra.

**BRCA RNA-seq data for orthogonal evidence**. We search orthogonal evidence of BRCA peptides in BRCA RNA-seq. A Python script (https://github.com/yafeng/proteogenomics_python/scam_bams.py) searches the number of reads that overlap (minimum 1 bp) the genomic region of novel peptides from BAM files (maximum mismatch =1). This script reads the GFF3 format file of identified peptides and BAM files as input files and outputs a count table. The RNA-seq BAM files (63 BRCA samples and 10 normal adjacent tissues) were downloaded from TCGA (Supplementary Data 1, Table 7)[54]. The GFF3 files of all BRCA datasets were merged into one.

**Quantification of novel coding loci**. For label-free quantification, the identified novel peptides were quantified by extracting MS1 maximum peak intensity using moFF[22]. Specifically, for the CRC dataset (PXD002137) which used label-free quantification, the log2 MS1 intensity of peptides belonging to a particular novel coding locus was summed, and paired *t* test was performed for matched samples. For other datasets based on TMT relative quantification, the peptide log2 ratio was calculated using internal reference as the denominator and was summarized into novel loci log2 ratio by taking the median value. To compare expression levels between tumors and normal tissues (unmatched samples), a two-sided *t* test was performed.

**Neoantigen prediction**. The NetCTLpan server(http://www.cbs.dtu.dk/services/NetCTLpan/) integrates prediction of peptide MHC class I binding, proteasomal C terminal cleavage, and TAP transport efficiency[39]. The predicted values were calculated as the weighted average of the MHC, TAP, and C-terminal cleavage scores, and as %-Rank to a set of 1000 random natural 9mer peptides. The prediction parameters of NetCTLpan are set as follows: the sorting threshold ≤0.5% (%Rank of prediction score to a set of 1000 random natural 9mer peptides), the weight proportion of c-terminal amino acid residues was 0.225, and the weight proportion of TAP transport efficiency was 0.025.

**Reporting summary**. Further information on research design is available in the Nature Research Reporting Summary linked to this article.

## Data availability
This study is based on publicly accessible datasets listed in Supplementary Data 1. Table 1 with accession identifiers and URLs provided in details.

## Code availability
The pipeline, scripts, and manual are posted at Github (https://github.com/yafeng/pan-cancer-proteogenomics-analysis).

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

## Acknowledgements
This work was supported by grants from the National Key Research and Development Program of China (No. 2017YFC1311000); GuangDong Science and Technology Department (No. 2017B030314026); Science, Technology and Innovation Commission of

Shenzhen Municipality (No. JCYJ20160531193931852); Science and Technology Key Project of Guangdong Province, China (No. 2019B020229002); Guangdong Enterprise Key Laboratory of Human Disease Genomics (2020B1212070028). We are thankful that the Clinical Proteomic Tumor Analysis Consortium (NCI/NIH) for sharing the data and ProteomeXchange Consortium for providing an open access proteomics data repository. We would like to thank China National GeneBank (https://db.cngb.org), Shenzhen, China for providing computing resources that support the data analysis for the paper.

## Author contributions

Y.F.Z. conceived and initiated the project. Y.F.Z. built the analysis workflow. R.X. annotated the datasets, R.X., Y.F.Z., L.M., M.Y., Z.Z., X.C., F.J., and F.X. performed the analysis, Y.F.Z. and R.X. made the figures and drafted the paper together. Y.M.Z., F.L., and K.W. contributed to the interpretation of data and revising the paper. All author contributed finalizing the paper.

## Competing interests

The authors declare no competing interests.
