## [Peer Review File · Communications Biology]

Reviewers' comments:

Reviewer #1 (Remarks to the Author):

Xiang et al. report here an extensive proteogenomic analysis aimed to discover peptides deriving from the translation of atypical transcripts such as pseudogenes, lncRNAs or LINEs. The existence of non-canonical transcripts (such as non-coding RNA or pseudogenes) translation in various normal and cancerous tissues is now a well-established phenomenon [PMID 24870542, 32139545 and 31340039]. The novelty of the study is therefore limited. However, as claimed by the authors, few or no studies (to my knowledge) have reported a parallel comparison of normal vs cancer tissues for the expression (at protein level) of such non-canonical genes. The present study is therefore of undeniable conceptual interest. In particular, the use of large scale and transversal analyses on robust MS datasets is appreciable. Nevertheless, I have major concerns regarding technical and analytical issues all along the article. These issues question the validity of the conclusions and need to be addressed before any possibility of publication.

General comments:

1) The overall poor English of the article makes sometimes the comprehension very difficult. Lines 73-75, 92-94, 131-132 and 186-187 illustrate the problem. The whole article should be checked for grammar.

2) The article would benefit from showing more data. The absence of detailed tables with lists of novel peptides (with annotations and the various classifications attributed by the authors, see also my specific comments below) makes the evaluation of the validity of the results very difficult. It also undermines the interest of the paper. As example, for figure 1, a detailed table with the list of identified novel peptides, their parental annotation (e.g. pseudogene / lncRNA and the gene IDs), their genomic position (exon / intron / UTR or intergenic) and the samples in which they have been identified would be highly interesting. For figure 2, there is only one panel for 43 lines of text in results, adding at least one panel supporting the results at lines 134-137 would be desirable.

3) The article would benefit from a discussion comparing the results reported here with those reported by other key studies on the cryptic proteome (such as PMID 24870542, 32139545 and 31340039), especially regarding the proportion of novel peptides deriving from lncRNA vs pseudogenes (see my comments below).

Specific comments:

4) The venn diagram in figure 1a suggests the existence of novel peptides either common or not between normal and cancer tissues. However, I wonder whether comparing mass spectrometry results reported by studies having used different instruments and having prepared the samples in different ways would give reliable results. Could this difference of data origin influence the results showed in this venn diagram? This concern is notably supported by authors observation that "Among the downloaded datasets, CPTAC studies generated more pseudogene identifications, likely a result of high-quality data and larger cohort size." (line 132). Therefore, there are differences of data quality between the different studies that influence the identifications made by the authors.

I would rather suggest using the cancer samples and their matched normal tissues that the authors

used in figure 3 (which were reported within the same study and have same methods of analysis, preparation and similar numbers of replicates) to demonstrate the existence of normal or cancer-specific novel peptides.

5) In figure 1c, it is unclear what the different categories refer to. Pseudogenes and lncRNAs are also composed of exons and introns (while their exons are assumed not to code for amino acids). This plot therefore suggests that ~80% of novel peptides derive from intergenic regions, and therefore unannotated regions that should not be present in the MS databases used to perform the peptides identifications. If a peptide derives from the translation of the exon of a lncRNA, its genomic position is "exon" and not "intergenic" (if the lncRNA is in an intergenic region). Please clarify.

6) In figure 1d, were all pseudogenes classified in these 8 categories or were there others that did not match these categories and were not reported in that figure? How many pseudogenes in total did the authors have for normal and cancer? Please provide the full list of the identified pseudogenes with their classification in supplementary files.

7) Relating to line 106 ("Our analysis identified three groups of pseudogene expression: ubiquitous, non-specific and tissue specific expression (Figure 1e)."), please either provide a list of the pseudogenes with their classification (ubiquitous, non-specific or tissue specific) and their tissue expression or clearly show these classifications on figure 1e.

8) Figure 2: please indicate a name for y axis and unambiguously indicate the comparisons performed in matrices (indicate cancer identity vertically). There is a major discrepancy between what is indicated in the figure ("Overlap of pseudogene detected within cancer type") and the figure legend ("In the matrix, the number in the last column of each row indicate the number of novel coding loci"), is it pseudogenes or novel coding loci numbers that are reported? Please clarify and separate the main plot and the matrices into two different panels to disambiguate the figure.

9) Relating to figure 2, the overlap of novel transcripts is relatively low (~10-30%) between cancer datasets in matrices. As this low overlap could be due to technical differences in data acquisition between studies, it would be appropriate here to report the inter-patient sharing of novel peptides within each dataset (e.g. within CPTAC Phase_II_Data_S026 which has a high number of patients) in addition to the comparisons already performed by the authors.

10) Regarding line 134 (Among... ribosomal proteins), can the authors show the data supporting their claim? What is the extent of the difference of sharing between pseudogenes homologous to house-keeping genes vs the others?

11) The proportion of novel peptides deriving from lncRNAs (8 according to Table 3) seems to be very low compared to those deriving from pseudogenes (867 only in breast cancer according to figure 2). Could the authors discuss this impressive difference? Is it expected in light of previously published studies?

12) Figure 3 and 5: indicate in legend what are "QC" and "NCI7". In figure 3 replace "peptiders" by "peptides".

13) The conclusion that RHOXF1P3 pseudogene is associated with estrogen receptor signaling is only supported by the fact that this pseudogene is mainly expressed by ER positive cancers. However, the majority (~80%) of breast cancers are typically ER+, and given the low frequency of the RHOXF1P3 expression (~10%), this selective presence in ER+ breast cancers could be only due to luck. This association would need to be at least demonstrated by a fisher's exact test. The association is even

more doubtful given that the frequency of the RHOXF1P3 expression in the confirmation cohort (fig 4c) is identical between cancer and normal samples (9.7% vs 11%). The same comments also apply to figure 4d-e. Demonstrating this association requires more robust statistical validation, comparison with cancers or tissues non-responsive to estrogen signaling and would also need a biologically sound explanation.

14) Regarding lines 236-240, the authors should consider the recent findings reported in PMID 32345368. LINE expression (and translation) in normal tissues is actually widespread and much more common than previously thought.

15) In figure 6, the MHC binding peptides have been generated from novel peptides only present in cancer samples, as explained at line 255. Please clarify from which cancer samples these peptides were obtained (all those reported in the article or some of them?) and which normal samples were considered (those of figure 1 + the normal matched samples of figures 3-5?).

16) Relating to figure 6, the authors use the RNA expression of the transcripts coding for their novel peptides as proxy in GTEx data to conclude that they are possible targets for immunotherapy. However, they have not demonstrated that there is a correlation between RNA expression and peptide level for their novel peptides. Low levels of RNA do not guarantee low levels of peptide (as lowly abundant RNA could be highly translated). To support the tumor specificity of their targets, they need at least to evidence a significant positive correlation between RNA and peptide level in their cancer samples. They could also report in fig 6c the tpm expression of the transcripts in their tumor samples in comparison to their RNA expression in GTEx. Please indicate in this figure (6c) what is reported: average or median of tpm?

Reviewer #2 (Remarks to the Author):

This manuscript details the use of a proteogenomics analysis workflow to find and analyze peptides encoded by non-coding sequences in publicly available datasets comprising five different cancer types and 31 healthy human tissues. The authors describe the protein expression of several pseudogenes and lncRNAs, including their tumor specificity (or lack thereof), and give several examples of how their data expands on previous studies of particular pseudogenes and lncRNAs. They also predict a quantity of potential neoantigens translated from tumor-specific noncoding regions that could be used in the future as bases for cancer vaccines. The questions addressed in the manuscript are fascinating and the authors present a wealth of good data and analysis. The manuscript would benefit from a more consistent application of statistical analysis to bolster claims of significant results. In addition, the use of normal tissue as a point of comparison needs to be more

rigorous.

MAJOR COMMENTS:

1. The biggest concern I have with this manuscript is the size of the normal datasets. For the most part, each “normal tissue” here refers to a single sample, so it cannot rightly be said to be widely representative of normal tissue. I would encourage the authors to make use of the CPTAC matched normals to increase the size of their dataset everywhere possible (or, if they did so, to be more clear about that fact). This should also be addressed as a limitation in the Discussion section.
2. Along similar lines, I would like to see normal tissue used as a point of comparison in a couple of the analyses, specifically those done in Figure 2 (addressed in Results point 13 below) and Tables 2 and 3 (points 14-16). Again, please take advantage of CPTAC datasets where possible to gather a more diverse set of normal tissue.
3. There are several points in the paper where results are presented without any statistical analysis or measures of significance. I have pointed them out below, specifically points 4, 21, 22, 25, and 26 in the Results section.
4. There are a couple of claims made in the paper that need more support (points 17, 23, and 24).

OTHER COMMENTS:

Overall:

1. A copy editor would help improve the readability and clarity of the paper.
2. There are a lot of abbreviations used here, so a key to the abbreviations used would be helpful.
3. I was unable to find legends for the supplementary material.
4. The data produced by this study – the tables of non-coding sequence–derived peptides – is an excellent resource! Mention that fact somewhere – it’s an important contribution of this work and should be addressed.

Introduction:

1. “It is conventionally believed...deleterious mutations” (line 40-41) – It would be best to cite a source here.

Results:

1. I would like to see a little bit more of a description of the sources of the healthy tissue – I had to go to the supplement of the original paper to see things like gender and age of the patients. This could easily go in Table S1. Also, cite PRIDE database and CPTAC data portal here, and list the five cancer types – readers won’t want to have to go to a supplementary table to see what types of cancer are being covered here. Consider replacing one of the main tables in the manuscript with a table summarizing the data used (at minimum, cancer types, normal tissue types, number of samples of each).
2. Consider moving Figure S1 to the main manuscript and one of the tables (perhaps Table 1 – the data in Table 1 is highlighted in Table S2) into the supplement. The pipeline, while not the main focus of the paper, is something I imagine readers would like to be able to easily refer to.
3. In Figure 1B, and other figures in the manuscript, samples could be labeled in a more easily readable way. For instance, “COAD_s37_PNNL” is not especially informative, especially since “PNNL” is not referred to anywhere else in the manuscript. (Maybe something like “COAD (CPTAC) Sample 37”?) Generally, it is best to avoid using abbreviations that are never defined.
4. “This result indicates...genomic regions” (line 90-91) – A significance measure would be good here.
5. “Our further analysis discovered...derived from pseudogenes” (line 92-94) – Please go into a little more detail. Can be as simple as the name of the software tool used to determine this.

6. How were the samples shown in Figure S2A chosen?
7. "Previous proteogenomic studies have rarely performed quantitative analysis" (line 103) – Please be more specific as to the type of proteogenomic studies referred to here (the world of proteogenomics is vast and there are tons of quantitative analyses out there), and perhaps cite one or two that have done quantitative analysis.
8. Suggestion for Figure 1E: separate the three groups (ubiquitous/nonspecific/tissue specific) somehow.
9. It would be interesting to know the specific proportions of the three groups.
10. If you take the suggestion in #2 above, Table 1 can either be moved to its own sheet within Table S2, or you can point out that the representative pseudogenes/lncRNAs are highlighted in Table S2.
11. "We detected two...eight unique peptides)." (line 112-116) – The reader can figure out that this means exclusively normal tissue, but it might be helpful just to mention that specifically.
12. Suggestion for Figure 2: split into two panels – one for the bar graph and one for the matrices. This will allow for more specific references to the two parts of the figure.
13. I would be more interested to see a version of Figure 2 with the ubiquitous pseudogenes removed – it makes sense that the ubiquitous pseudogenes would overlap, but what pseudogenes are specific to one or more cancer types as opposed to normal tissue? Along those lines, what happens if you remove the pseudogenes that are tissue-specific or non-specific to the normal tissue that gives rise to the particular cancer type (e.g. if you only look at BRCA pseudogenes that are not also normal breast-specific pseudogenes)?
14. For peptides that are "recurrently detected", does this mean they were detected in multiple samples of each cancer type, or just in one or two samples of each cancer dataset? If the latter, I'm not sure it can be said that they're cancer-specific rather than patient-specific.
15. Are the recurrently detected pseudogenes in Table 2 also detected in their analogous normal tissue? For instance, is RHOXF1P3 just a pseudogene that is translated in breast and ovarian tissue, or is it specific to breast and ovarian cancer?
16. The questions in 14 and 15 also apply to Table 3.
17. I don't think the claim in lines 152-153 that DGCR5 probably encodes a functional protein product is supported by the data.
18. A small change in line 170 – I would change "probably" to "may". The reason for this is that there is a distinct possibility that there actually is a protein product of ATP8A2P1, but it was unable to be detected by MS.
19. I strongly suggest not imputing missing values in the COAD peptide abundance data (line 179-180). Instead, leave the values missing, or remove those rows. This keeps the data as accurate as possible.
20. The tab in Table S3 is mislabeled – should say it refers to Figure 3A.
21. A paired t-test for significance should be performed and reported for the data in Figure 3B. Also, using a box-and-whisker plot to present data with matched samples is a bit misleading, as it implies that the data points are independent – see Figure 2 in <https://doi.org/10.1371/journal.pbio.1002128> for ideas on presenting this in a more informative way.
22. Again, please report significance for the data in Figures 3C-E.
23. "This contradicts...normal tissues" (line 189-190) I'm not sure this is true. I looked at the paper cited here and it appears that they also show increased expression of DCGR5 RNA in cancer as compared to normal tissue.
24. I think the paragraph about RHOXF1P3 (line 198-209) needs more support. For instance, the confirmatory set has roughly the same proportion of RHOXF1P3-expressed samples in the normal and tumor sets. Also, what are the ER statuses of the samples in the ovarian and breast confirmatory sets?

25. Significance value in Figure 4F?
26. Make sure to reference Figure 5 somewhere in the manuscript, and add significance values.
27. Line 256: should reference Table S5, not Table S6.
28. For readers unfamiliar with parameters used in NetMHCpan and NetCTLpan, please explain what the thresholds mentioned in line 258 refer to.
29. Figure 6B is difficult to interpret, and may be better off either replaced with a simple Venn diagram (overlap between MHC and CTL) or moved to the supplement (or both).
30. Table S5, as cited in line 261, doesn't really lay out the overlap – maybe add a third sheet to that Excel file that contains only the overlap novel loci?
31. How were the antigen peptides shown in Figure 6C chosen?

Discussion:

1. "According to...non-coding regions" (line 294-296) – I'd suggest moving this to the next paragraph.
2. "Our results suggest...normal and tumors" (line 298-299) – Please elaborate on how the results suggest this.

Reviewer #3 (Remarks to the Author):

The authors studied published proteomics data from 926 cancer samples of five cancer types and 31 different healthy human tissues to describe the translation of pseudogenes. Although I find this work relevant and contributes to the still very unknown field of pseudogenes translation I can not recommend it for publication until the authors work on the following aspects:

- What is the statistical criteria to choose 31 healthy human tissues? Can the authors describe further the criteria the used for the selection of those matched 31 tissues?
- The authors suggest that there is a common mechanism that regulates the translation of pseudogenes in healthy and tumor tissues, especially those ones involved in cytoskeleton. How can the authors link this possible mechanism to the upregulation of many cytoskeleton proteins in several cancer types?
- The authors describe throughout the paper examples of pseudogenes in specific cancers, RHOXF1P3, PA2G4P4, MCTS2P, DBCR5, and so on. However there is a very poor description of these genes and their products and no discussion at all how they might function in those specific cancers
- What is the rational of using the left-shifted Gaussian distribution method to carry out imputations? What is the percentage of missing values? Did the authors try other imputation methods to address low abundance?
- Can the authors support the findings of RHOXF1P3 expression in ER positive tumors with statistical evidence on the main text?
- The authors do not describe any difficulty/challenge/adjustements in the performance of the proteogenomics workflow they used. What is that the case?

Minor aspects:

- space after Z. on line 12
- The current format of supplementary tables does not help to retrieve information intuitively. For example, What is the meaning of the coloring used in the middle to supplementary table S2?
- Line 56, write the reference properly
- Missing space before 3 on line 249
- Can you rephrase "genomics data detected mutations" on line 319. What does it mean?
- Can the authors describe in detail the quantification method of the proteomic datasets?
- Can the authors describe more in detail the NetCTLpan parameters and explain why they choose those parameters?
- Poor format of the tables in the main text
- Correct know peptides by known peptides in Figure S1

Dear reviewers,

Thank you very much for the advice on correcting my manuscript entitled **“Proteogenomics analysis of non-coding region encoded peptides in normal tissues and five cancer types”**. And we revised the manuscript thoroughly based on the comments. In the revised manuscript, all the suggestions and comments have been taken into consideration carefully. Here are point-to-point answers to the questions raised by reviewers.

Reviewer #1 (Remarks to the Author):

Xiang et al. report here an extensive proteogenomic analysis aimed to discover peptides deriving from the translation of atypical transcripts such as pseudogenes, lncRNAs or LINEs. The existence of non-canonical transcripts (such as non-coding RNA or pseudogenes) translation in various normal and cancerous tissues is now a well-established phenomenon [PMID 24870542, 32139545 and 31340039]. The novelty of the study is therefore limited. However, as claimed by the authors, few or no studies (to my knowledge) have reported a parallel comparison of normal vs cancer tissues for the expression (at protein level) of such non-canonical genes. The present study is therefore of undeniable conceptual interest. In particular, the use of large scale and transversal analyses on robust MS datasets is appreciable. Nevertheless, I have major concerns regarding technical and analytical issues all along the article. These issues question the validity of the conclusions and need to be addressed before any possibility of publication.

Revision: Thanks for the positive comments. In the revised manuscript, all the major concerns have been taken into consideration carefully.

General comments:

1) The overall poor English of the article makes sometimes the comprehension very difficult. Lines 73-75, 92-94, 131-132 and 186-187 illustrate the problem. The whole article should be checked for grammar.

1-1) Revision: Thanks for the good advice. We reorganized or revised these sentences.

Lines 73-75. These sentences that “As for different cancer datasets, genomics data detected mutant proteins’ sequences are supplemented to the core search database (See details in Method)” in the first manuscript were changed into “As for different cancer datasets, a collection of cancer mutations were downloaded from the Cancer Genomic Data Server (18). These mutations were then converted to mutant protein sequences using customized scripts and supplemented to the search database of corresponding cancer type. (See details in Method)”

Lines 92-94. In revised manuscript, we clarified genomic positions of the novel peptides in revised figure 1d legend.

Pseudogene: all categories of pseudogene (if novel peptide belongs to pseudogene, we will not count it again in other categories).

lncRNA: ncRNA

Exonic: coding gene's exon, not at canonical reading frame

Intronic: coding gene's intron

Intronic-exonic boundary: peptide spanning over coding gene's exon-intron boundary

UTR region: Untranslated region of coding gene

Upstream: Upstream of coding gene (1kb distance to closest UTR)

Downstream: Downstream_of coding gene (1kb distance to closest UTR)

Figure 1d

These sentences that “Next, we categorized the novel-coding loci based on their genomic positions (Figure 1c). This result indicates tumor and normal has no major differences of expressed novel coding loci in terms of genomic regions. **Our further analysis discovered the majority of novel coding loci in intergenic and intronic regions are derived from pseudogenes.** We therefore annotated the parental genes’ functions of translated pseudogenes (Figure 1d). In consistent to the findings revealed by previous RNA-seq data analysis⁶, the frequently detected pseudogenes in healthy tissues and tumors are homologous to house-keeping genes such as cytoskeleton proteins (actin, keratin,98 tubulin), ribosomal proteins, nuclear ribonucleoproteins, heat shock proteins and eukaryotic translation elongation factor, peptidylprolyl isomerase (Figure 1d, supplementary figure S2 a). Our results suggest that there is a common mechanism that regulates the translation of pseudogenes in healthy and tumor tissues” in the old manuscript were changed into “Next, we annotated the identified novel peptides detected from the healthy tissue dataset and cancer datasets based on

their origin (Figure 1d). In consistent with the findings in Kim *et al.*'s study (26) and our previous work (4), **the majority of novel peptides are from translating pseudogenes**. LncRNA is the second major source of identified novel peptides. In addition, many novel peptides were identified from retroelements (Long interspersed element-1, LINE-1) and intronic regions in cancer datasets.”

Lines 131-132. In the old manuscript, figure 2 was difficult to understand, so we removed it in revised manuscript and made a new figure 3 to state “Overlap of detected non-coding gene translation in different samples and datasets”. Therefore, we deleted Lines 131-132.

These sentences that “We analyzed the overlap of detected novel coding loci in different samples with in each study in figure 3a. The novel coding loci are divided into four groups by the percentage of samples they were identified. For example, the dataset PXD002619 produced the largest number of novel coding loci, but two third were identified in less than 25% samples. On average, one third of all novel loci were identified in more than 50% of samples.

Among different cancer datasets (in total 13 datasets covering five cancer types), 46% of pseudogene identifications were repeatedly detected in at least two datasets. In comparison, only 16% of non-pseudogenes were detected in more than one dataset. Further analysis showed that 93% of pseudogenes identified commonly in 8 to 13 different datasets belong to housekeeping genes, which suggest pseudogenes derived from house-keeping genes are also ubiquitously expressed in different cancer types. (Figure 3b)” were added into revised manuscript.

Figure 3

a) the overlap of detected novel coding loci in different samples

b) overlap of pseudogenes between datasets

overlap of non-pseudogenes between datasets

Legend: Figure 3. a) the overlap of detected novel coding loci in different samples. For example, red bar indicates the number of novel coding loci that are only missing in 0%-25% of the samples, while blue bar indicates the number of novel coding loci missing in 75%-100% of the samples. b) overlap of pseudogenes and non-pseudogenes between datasets. 54% of pseudogenes only appear in one dataset, and 40.8% pseudogenes appear in 2 to 7 data sets, and 5.2% pseudogenes appear in 8 to 13 data sets. While 97% of pseudogenes appearing in 8 to 13 data sets belong to housekeeping genes. 84.4% of non-pseudogenes only appear in one data set with higher specificity.

Lines 186-187. These sentences that “In other cancer datasets, we also detected several non-coding gene encoded peptides show increased expression in tumors” in the old manuscript were changed into “In addition to pseudogenes, we also found several long

non-coding RNA encoded peptides were detected in specific cancers”

2) The article would benefit from showing more data. The absence of detailed tables with lists of novel peptides (with annotations and the various classifications attributed by the authors, see also my specific comments below) makes the evaluation of the validity of the results very difficult. It also undermines the interest of the paper. As example, for figure 1, a detailed table with the list of identified novel peptides, their parental annotation (e.g. pseudogene / lncRNA and the gene IDs), their genomic position (exon / intron / UTR or intergenic) and the samples in which they have been identified would be highly interesting. For figure 2, there is only one panel for 43 lines of text in results, adding at least one panel supporting the results at lines 134-137 would be desirable.

1-2) Revision: Thanks for the good advice. We attached our detailed tables with lists of novel peptides in supplementary table S2 (identified novel peptides from healthy tissues) and supplementary table S3 (identified novel peptides from cancer tissues) in revised manuscript.

For figure 2, figure 2 was difficult to understand, so we removed it in revise manuscript and made a new figure 3 to state “Overlap of detected non-coding gene translation in different samples and datasets”. see the responses to 1-1) for details.

lines 134-137. These sentences that “Among different cancer types, the repeatedly detected pseudogenes are those homologous to house-keeping genes including pseudogenes of eukaryotic elongation factors, GAPDH, actin/keratin/tubulin, heat shock protein family, heterogeneous nuclear ribonucleoprotein and ribosomal proteins” were changed to “Among different cancer datasets (in total 13 datasets covering five cancer types), 46% of pseudogene identifications were repeatedly detected in at least two datasets. In comparison, only 16% of non-pseudogenes were detected in more than one dataset. Further analysis showed that 93% of pseudogenes that were identified commonly in 8 to 13 different datasets belong to housekeeping genes, which suggest pseudogenes derived from house-keeping genes are also ubiquitously expressed in different cancer types. (Figure 3b).”

Figure 3b

Legend: 3b) overlap of pseudogenes and non-pseudogenes between datasets. 54% of pseudogenes only appear in one dataset, and 40.8% pseudogenes appear in 2 to 7 data sets, and 5.2% pseudogenes appear in 8 to 13 data sets. While 97% of pseudogenes appearing in 8 to 13 data sets belong to housekeeping genes. 84.4% of non-pseudogenes only appear in one data set with higher specificity.

3) The article would benefit from a discussion comparing the results reported here with those reported by other key studies on the cryptic proteome (such as PMID 24870542, 32139545 and 31340039), especially regarding the proportion of novel peptides deriving from lncRNA vs pseudogenes (see my comments below).

1-3) Revision: Thanks for the good advice. According to reviewer's suggestion, we compared our results with other two cryptic proteomes (PMID 32139545 and 31340039, PMID 24870542 was not compared due to a lack of complete novel ORF sequence). These compared results that "We compared our proteomics results with two recent studies that used ribosomal profiling and full-length mRNA sequencing to search translated non-coding genes in multiple cell types and cancer cell lines (30, 31). Lu *et al.* identified 2969 translating non-coding gene from mRNA sequencing and ribosomal profiling data, and detected 10% (308) non-coding gene encoded new proteins (372 unique peptides) with shotgun proteomics. Among these new proteins, 59 were also identified in our results (See supplementary table S4). These include MCTS2P, MKKS

5' UTR ORF, LINE-1 ORF1 and PA2G4P4. In comparison, only eight novel CDS were found in common between Chen *et al.*'s study (30) and ours, which could be due to the sample difference since their novel CDS were identified from induced pluripotent stem cells (iPSCs), iPSCs derived cardiomyocytes and human foreskin fibroblasts. Of note, these common novel CDS include MCTS2P, STARD10 5' UTR ORF and TSGA10 5' UTR ORF" were added into revised manuscript.

Specific comments:

4) The venn diagram in figure 1a suggests the existence of novel peptides either common or not between normal and cancer tissues. However, I wonder whether comparing mass spectrometry results reported by studies having used different instruments and having prepared the samples in different ways would give reliable results. Could this difference of data origin influence the results showed in this venn diagram? This concern is notably supported by authors observation that "Among the downloaded datasets, CPTAC studies generated more pseudogene identifications, likely a result of high-quality data and larger cohort size." (line 132). Therefore, there are differences of data quality between the different studies that influence the identifications made by the authors.

I would rather suggest using the cancer samples and their matched normal tissues that the authors used in figure 3 (which were reported within the same study and have same methods of analysis, preparation and similar numbers of replicates) to demonstrate the existence of normal or cancer-specific novel peptides.

1-4) Revision: Thanks for the good advice. As you said, the difference of data origin influenced the venn diagram. The venn diagram in figure 1a showed that how many novel peptides were detected in tumor or normal tissues, and how many overlapping peptides there are. The result was not used to define cancer specific novel peptides.

Based on your suggestion, we have revised our filtering criteria for selecting neoantigen candidates as:

“1) The average expression of novel loci in tumor tissue was up-regulated by 1.5 times compared with matched normal tissue (restricted to the datasets in which matched normal tissues are available).2) Peptides were supported by NetCTLpan predictions.”

5) In figure 1c, it is unclear what the different categories refer to. Pseudogenes and lncRNAs are also composed of exons and introns (while their exons are assumed not to code for amino acids). This plot therefore suggests that ~80% of novel peptides derive from intergenic regions, and therefore unannotated regions that should not be present in the MS databases used to perform the peptides identifications. If a peptide derives from the translation of the exon of a lncRNA, its genomic position is “exon” and not “intergenic” (if the lncRNA is in an intergenic region). Please clarify.

1-5) Revision: Thanks for the good advice. We clarified the categories in revised manuscript. If a peptide derives from the translation of the exon of a lncRNA, its genomic position is lncRNA. Exons and introns refer to coding genes' regions (See “Revision 1-1, Lines 92-94” for details)

6) In figure 1d, were all pseudogenes classified in these 8 categories or were there others that did not match these categories and were not reported in that figure? How many pseudogenes in total did the authors have for normal and cancer? Please provide the full list of the identified pseudogenes with their classification in supplementary files.

1-6) Revision: Thanks for the good advice. Pseudogenes which did not match these categories were not reported in figure 1d. “The eight major categories include 426 and 2044 novel peptides, comprising 70% and 84% total novel peptides detected from the healthy tissues and cancer datasets.”

We made a new supplementary file (table S4) for providing the full list of the identified

pseudogenes with their descriptions.

7) Relating to line 106 (“Our analysis identified three groups of pseudogene expression: ubiquitous, non-specific and tissue specific expression (Figure 1e).”), please either provide a list of the pseudogenes with their classification (ubiquitous, non-specific or tissue specific) and their tissue expression or clearly show these classifications on figure 1e.

1-7) Revision: Thanks for the good advice. We marked ubiquitous, non-specific and tissue specific in figure 1e (now move to figure 2). We described it with “Our analysis identified three groups of novel coding loci expression: 12 ubiquitous (expressed in at least 15 tissues), 93 non-specific (expressed in 2 to 15 tissues, robustly translated in one or two tissues but frequently translated at lower levels in other tissues.), and 114 tissue-specific expression (expressed in one tissue) (Figure 2).” in revised manuscript.

Legend: Figure 2. Heatmap of MS1 intensity of non-coding gene encoded peptides detected in normal tissues.

8) Figure 2: please indicate a name for y axis and unambiguously indicate the

comparisons performed in matrices (indicate cancer identity vertically). There is a major discrepancy between what is indicated in the figure (“Overlap of pseudogene detected within cancer type”) and the figure legend (“In the matrix, the number in the last column of each row indicate the number of novel coding loci”), is it pseudogenes or novel coding loci numbers that are reported? Please clarify and separate the main plot and the matrices into two different panels to disambiguate the figure.

1-8) Revision: Thanks for the good advice. We discard the old figure 2, because the bar plot is too complicated to interpret and inter-datasets overlap is confounded by different instrumentations and methods. We take the suggestion to compare the overlap of detected non-coding gene translation in different samples within each study. We made a new plot in figure 3 to illustrate it.

Along with the new figure 3, we added the following sentences in the manuscript.

“we analyzed the overlap of detected novel coding loci in different samples within each study in figure 3a. The novel coding loci are divided into four groups by the percentage of samples they were identified. For example, the dataset PXD002619 produced the largest number of novel coding loci, but two third were identified in less than 25% samples. On average, one third of all novel loci were identified in more than 50% of samples.

Among different cancer datasets (in total 13 datasets covering five cancer types), 46% of pseudogene identifications were repeatedly detected in at least two datasets. In comparison, only 16% of non-pseudogenes were detected in more than one dataset. Further analysis showed that 93% of pseudogenes identified commonly in 8 to 13 different datasets belong to housekeeping genes, which suggest pseudogenes derived from house-keeping genes are also ubiquitously expressed in different cancer types. (Figure 3b)”.

Figure 3

a) the overlap of detected novel coding loci in different samples

b) overlap of pseudogenes between datasets

overlap of non-pseudogenes between datasets

Legend: Figure 3. a) the overlap of detected novel coding loci in different samples. For example, red bar indicates the number of novel coding loci that are only missing in 0%-25% of the samples, while blue bar indicates the number of novel coding loci missing in 75%-100% of the samples. b) overlap of pseudogenes and non-pseudogenes between datasets. 54% of pseudogenes only appear in one dataset, and 40.8% pseudogenes appear in 2 to 7 data sets, and 5.2% pseudogenes appear in 8 to 13 data sets. While 97% of pseudogenes appearing in 8 to 13 data sets belong to housekeeping genes. 84.4% of non-pseudogenes only appear in one data set with higher specificity.

9) Relating to figure 2, the overlap of novel transcripts is relatively low (~10-30%) between cancer datasets in matrices. As this low overlap could be due to technical

differences in data acquisition between studies, it would be appropriate here to report the inter-patient sharing of novel peptides within each dataset (e.g. within CPTAC Phase_II_Data_S026 which has a high number of patients) in addition to the comparisons already performed by the authors.

1-9) Revision: Thanks for the good advice. We agree that Inter-datasets overlap is also related to instrumentation and methods. According to reviewer's good advice, we added a new figure 3 to illustrate the overlap of detected non-coding gene translation in different samples and datasets.

See the responses to 1-8) for details. Thanks.

10) Regarding line 134 (Among... ribosomal proteins), can the authors show the data supporting their claim? What is the extent of the difference of sharing between pseudogenes homologous to house-keeping genes vs the others?

1-10) Revision: Thanks for the good advice. We made figure 3b to support the repeatedly detected pseudogenes are homologous to house-keeping genes. "Among different cancer datasets (in total 13 datasets covering five cancer types), 46% of pseudogene identifications were repeatedly detected in at least two datasets. In comparison, only 16% of non-pseudogenes were detected in more than one dataset. Further analysis showed that 93% of pseudogenes identified commonly in 8 to 13 different datasets belong to housekeeping genes, which suggest pseudogenes derived from house-keeping genes are also ubiquitously expressed in different cancer types. (Figure 3b)".

See the responses to 1-8) for details. Thanks.

11) The proportion of novel peptides deriving from lncRNAs (8 according to Table 3) seems to be very low compared to those deriving from pseudogenes (867 only in breast cancer according to figure 2). Could the authors discuss this impressive difference? Is it expected in light of previously published studies?

1-11) Revision: Thanks for the good advice. It is similar to the Kim *et al.*'s study [PMID 24870542], which pseudogenes accounts for 72% of novel protein-coding regions (140/193) and ncRNAs accounts for 4% of novel protein-coding regions(9/193) (see below).

	Number of genes involved or annotation confirmed /added/altered
Genes whose products were detected	17,294
Confirmed exons	223,385
Confirmed N termini	4,105
Confirmed exon-exon junctions	66,947
Signal peptide cleavage site	329
Confirmation of annotated sites	128
Unannotated cleavage sites	201
Novel protein-coding regions	193
Pseudogenes	140
Non-coding RNAs	9
Upstream ORFs	29
Other ORFs	15
Novel coding regions/exons	106
Novel N termini	198
Gene/protein extensions	70
N-terminal extension	58
C-terminal extension	12
Exon extension	40

Legend: Overall summary of the results from Kim *et al.*'s study [PMID 24870542].

These sentences correspond to "Further, in consistent with the findings in Kim *et al.*'s study and our previous work, the majority of novel peptides are from translating pseudogenes. LncRNA is the second major source of identified novel peptides. The low percentage of novel peptides detected from lncRNAs is in line with previous study by Guttman et al, in which a comprehensive analysis of ribosomal profiling data provided supporting evidence that the large majority of lncRNAs do not encoded proteins." in revised manuscript.

12) Figure 3 and 5: indicate in legend what are "QC" and "NCI7". In figure 3 replace "peptiders" by "peptides".

1-12) Revision: Thanks for the good advice. Please note Old figure 3,4,5 moved to figure 4,5,6, separately. We added explanation in legend what are "QC" and "NCI7".

These sentences correspond to ““NCI7” and “QC” are both reference samples. The NCI7 reference sample consists of a mixture of lysate from cell lines NCI-H23, RPMI-8226, T47D, A549, COLO205, NCI-H226, CCRF-CEM. The QC sample is an independently obtained kidney tumor reference sample.” in revised figure 4 legend.

In figure 3 “peptiders” was replaced by “peptides” in revised manuscript.

13) The conclusion that RHOXF1P3 pseudogene is associated with estrogen receptor signaling is only supported by the fact that this pseudogene is mainly expressed by ER positive cancers. However, the majority (~80%) of breast cancers are typically ER+, and given the low frequency of the RHOXF1P3 expression (~10%), this selective presence in ER+ breast cancers could be only due to luck. This association would need to be at least demonstrated by a fisher’s exact test. The association is even more doubtful given that the frequency of the RHOXF1P3 expression in the confirmation cohort (fig 4c) is identical between cancer and normal samples (9.7% vs 11%). The same comments also apply to figure 4d-e. Demonstrating this association requires more robust statistical validation, comparison with cancers or tissues non-responsive to estrogen signaling and would also need a biologically sound explanation.

1-13) Revision: Thanks for the good advice. We tried a fisher’s exact test. The p-value is not significant. Indeed, more evidence is needed to prove that RHOXF1P3 pseudogene is associated with estrogen receptor signaling. We modified our statement. We **removed** that “Next, we tried to correlate the expression of non-coding gene encoded peptides with tumor subtypes. Interestingly, the pseudogene RHOXF1P3 detected with eight unique peptides showed increased expression (2 to 16-fold up-regulation) in a subset of estrogen receptor (ER) positive breast cancer from CPTAC breast cancer Discovery cohort (Figure 4a-4b)¹⁹. RHOXF1P3 peptides were also detected in CPTAC breast203 cancer confirmatory cohort, expressed in 13 out of 133 breast tumor and 2 out of 18 breast normal samples (Figure 4c). In addition, RHOXF1P3 encoded peptides were also detected in two ovarian cancer patients (Figure 4d). We then analyzed the expression of RHOXF1P3 in a published RNA-seq data including 63 breast tumors and 10 adjacent normal tissues (Figure 4e). These combined results indicate RHOXF1P3 is upregulated in breast tumors and it is likely to

be associated with ER signaling in breast and ovarian cancer.”

The text has been updated as the following. “In the two CPTAC breast cancer datasets, the pseudogene RHOXF1P3 was identified with eight and seven unique peptides respectively, covering 89% of amino acid sequences of the open reading frame encoded by this pseudogene (Figure 5a). More interestingly, the pseudogene RHOXF1P3 encoded peptides were up-regulated (2 to 16-fold) in a subset of breast cancer patients both in CPTAC breast cancer Discovery and Confirmatory cohort (Figure 5b,5c) (27). Besides, RHOXF1P3 encoded peptides were also detected in two ovarian cancer patients (Figure 5d). We then analyzed the expression of RHOXF1P3 in a published RNA-seq data including 63 breast tumors and 10 adjacent normal tissues, which also showed up-regulated expression of RHOXF1P3 in tumor samples (Figure 5e). Together, our results demonstrated that pseudogene RHOXF1P3 is not only translated, but also upregulated in a subset of breast tumors.”

Figure 5

Legend: Figure 5. Differential expressed non-coding gene encoded peptides in breast cancer. a) predicted protein sequence of pseudogene *RHOXF1P3* (amino acids

in read were supported by detected peptides). b) relative expression of *RHOXFIP3* encoded peptides in 77 breast tumors. Grey color boxes indicate missing values. c) relative expression of *RHOXFIP3* encoded peptides in 133 breast tumors and 18 adjacent normal tissues. d) relative expression of *RHOXFIP3* encoded peptides in 86 ovarian tumors and 22 normal ovarian tissues. QC: this is a peptide sample produced at JHU by digesting a large ovarian tumor; this sample was then aliquoted and used at both PNNL and JHUZ for demonstrating the reproducibility in the TMT-10 analysis of ovarian prospective samples. e) RNA seq read count of *RHOXFIP3* transcript in breast tumor and normal.

14) Regarding lines 236-240, the authors should consider the recent findings reported in PMID 32345368. LINE expression (and translation) in normal tissues is actually widespread and much more common than previously thought.

1-14) Revision: Thanks for the good advice. We discussed and cited paper PMID 32345368 in revised manuscript that “Surprisingly, we also detected peptides of LINE-1 ORF1 in healthy tissues, including lung, ovary and prostate (supplementary table S2). **Another independent study using RNA-sequencing data also observed widespread expression of retroelements in human somatic tissues (42).** This may be explained by a recent finding that LINE-1 activity becomes derepressed in senescent cells and healthy tissues could have senescent cells at old age (43).”

15) In figure 6, the MHC binding peptides have been generated from novel peptides only present in cancer samples, as explained at line 255. Please clarify from which cancer samples these peptides were obtained (all those reported in the article or some of them?) and which normal samples were considered (those of figure 1 + the normal matched samples of figures 3-5?).

1-15) Revision: Thanks for the good advice. The description was not clear. According to reviewer's good advice, we only selected novel peptides which have tumor and matched normal tissue. Therefore, we revised this part of predicting neoantigens. **We removed that** "The process of tumor neoantigen presentation based on the tumor

specific novel peptide mainly depends on: 1) whether the novel peptide can be selectively hydrolyzed by proteasome; 2) whether the digested peptide can be selectively transported by antigen processing-related transporters; 3) if the peptides can bind to MHC molecules, and can the presented antigen be recognized by T cells. Therefore, novel tumor-specific peptides were predicted by combining NetCTLpan and NetMHCpan, both intracellular presentation of MHC I class antigens and the ability of MHC and extracellular binding of antigens are considered^{26,27}.

After filtering out all novel peptides expressed in healthy tissues, 1587 peptides from 524 pseudogenes were used in neoantigen prediction (table S6). By NetMHCpan and NetCTLpan 4.0 prediction, 444 and 296 pseudogenes/lncRNAs have at least one 9-mer peptide with predicted affinity ranked at threshold $\leq 0.5\%$ and $\leq 1\%$, respectively (Figure 6a). Taking the overlap novel loci of the two softwares, 296 novel loci encode peptides predicted to be presented by at least one HLA allele and recognized by T cells (Figure 6b, supplementary table S5). Some examples of predicted neoantigens from non-coding genes are shown in Table 4. In GTEx-data, the transcripts that encode these 263 antigen peptides are poorly expressed or not expressed in healthy tissues (Figure 6c), allowing them to be explored as peptide vaccine to treat cancer⁹.

The text has been updated as the following: "T cells recognize and bind to a peptide–MHC complex is a complex process with many crucial steps. First, the abnormal proteins are hydrolyzed by proteases into peptide fragments in the cytoplasm, and then peptide fragments transported by the transporter associated with antigen processing (TAP) protein into the endoplasmic reticulum, where the peptide bind to an MHC molecule³⁷. The NetCTLpan server integrates predictions of proteasomal C terminal cleavage, TAP transport efficiency, and MHC class I binding affinities, which takes into account the antigen processing and presentation³⁸. Therefore, we used NetCTLpan server to predict the neoantigen.

We selected neoantigen candidates based on the following criteria: 1) The average expression of novel loci in tumor tissue was up-regulated by 1.5 times compared with matched normal tissue (restricted to the datasets in which matched normal tissues are available). 2) Peptides were supported by NetCTLpan predictions. By NetCTLpan

prediction, 64 pseudogenes/lncRNAs have at least one 9-mer peptide with predicted affinity ranked at threshold $\leq 0.5\%$ (supplementary table S6). These results suggest that there are a large number of candidate neoantigens in the non-coding regions. Several candidate neoantigens were listed table 1. Noteworthy, the 9-mer peptide (HEDTGNPGL) encoded by pseudogene RHOXF1P3 and the peptide (RLQEGLAAV) encoded by lncRNA lnc-SERPIND1-41:10 (DGCR9 intron) were predicted as neoantigens. "

Please note that the new version of NetCTLpan server uses a neural network to predict the binding of MHC peptides, including the function of the NetMHCpan server. Therefore, we discarded the prediction of NetMHCpan server in revised manuscript.

16) Relating to figure 6, the authors use the RNA expression of the transcripts coding for their novel peptides as proxy in GTEx data to conclude that they are possible targets for immunotherapy. However, they have not demonstrated that there is a correlation between RNA expression and peptide level for their novel peptides. Low levels of RNA do not guarantee low levels of peptide (as lowly abundant RNA could be highly translated). To support the tumor specificity of their targets, they need at least to evidence a significant positive correlation between RNA and peptide level in their cancer samples. They could also report in fig 6c the tpm expression of the transcripts in their tumor samples in comparison to their RNA expression in GTEx. Please indicate in this figure (6c) what is reported: average or median of tpm?

1-16) Revision: Thanks for the good advice. Taking together with other reviewers' comments, we decided to use only matched normal tissues to filter cancer specific novel peptides. We discarded the old figure 6 and have updated the neoantigen prediction analysis. See the responses to 1-15) for details.

Reviewer #2 (Remarks to the Author):

This manuscript details the use of a proteogenomics analysis workflow to find and analyze peptides encoded by non-coding sequences in publicly available datasets comprising five different cancer types and 31 healthy human tissues. The authors

describe the protein expression of several pseudogenes and lncRNAs, including their tumor specificity (or lack thereof), and give several examples of how their data expands on previous studies of particular pseudogenes and lncRNAs. They also predict a quantity of potential neoantigens translated from tumor-specific noncoding regions that could be used in the future as bases for cancer vaccines. The questions addressed in the manuscript are fascinating and the authors present a wealth of good data and analysis. The manuscript would benefit from a more consistent application of statistical analysis to bolster claims of significant results. In addition, the use of normal tissue as a point of comparison needs to be more rigorous.

Thanks for the positive comments, we have revised the manuscript based on your comments.

MAJOR COMMENTS:

1. The biggest concern I have with this manuscript is the size of the normal datasets. For the most part, each “normal tissue” here refers to a single sample, so it cannot rightly be said to be widely representative of normal tissue. I would encourage the authors to make use of the CPTAC matched normals to increase the size of their dataset everywhere possible (or, if they did so, to be more clear about that fact). This should also be addressed as a limitation in the Discussion section.

Thanks for the good advice. Our old description was not clear. There are two types of "normal" tissues, one is the 31 different tissues types (histologically healthy tissues), the other is matched normal tissues in cancer studies. We have now refer the former as "the 31 healthy tissues", the latter as "matched normal tissues".

We updated our filtering criteria for neoantigen prediction.

1) The average expression of novel loci in tumor tissue was up-regulated by 1.5 times compared with matched normal tissue (restricted to the datasets in which matched normal tissues are available). 2) Peptides were supported by NetCTLpan predictions. We acknowledge this limitation that the representation of normal tissues was not wide enough, we have added this limitation to discussion.

2. Along similar lines, I would like to see normal tissue used as a point of comparison in a couple of the analyses, specifically those done in Figure 2 (addressed in Results point 13 below) and Tables 2 and 3 (points 14-16). Again, please take advantage of CPTAC datasets where possible to gather a more diverse set of normal tissue.

We have removed the figure 2 due to its complication to interpret. Instead, we take one of reviewers' suggestion to summarize the inter-samples overlap of novel loci within each study (figure 3a).

Figure 3

a) the overlap of detected novel coding loci in different samples

b) overlap of pseudogenes between datasets

overlap of non-pseudogenes between datasets

1 2-7 8-13 housekeeping gene non-housekeeping gene

3. There are several points in the paper where results are presented without any statistical analysis or measures of significance. I have pointed them out below, specifically points 4, 21, 22, 25, and 26 in the Results section.

We have now added statistical test results in corresponding place.

4. There are a couple of claims made in the paper that need more support (points 17, 23, and 24).

See detailed responses below.

OTHER COMMENTS:

Overall:

1. A copy editor would help improve the readability and clarity of the paper.

2-1) Revision: Thanks for the good advice. We have worked on the language and improved the readability of the paper.

2. There are a lot of abbreviations used here, so a key to the abbreviations used would be helpful.

2-2) Revision: Thanks for the good advice. We made an abbreviations table “Table of non-standard abbreviations” in revised manuscript.

Table of non-standard abbreviations

Abbreviation	Full name
BRCA	Breast Cancer
CCRCC	Clear Cell Renal Cell Carcinoma
CRC	Colorectal cancer
CPTAC	The Clinical Proteomic Tumor Analysis Consortium
S22_CRC	CPTAC colorectal cancer study S022
iPSCs	induced pluripotent stem cells
lncRNAs	long non-coding RNAs
OV	Ovarian cancer
PRIDE	PRoteomics IDentifications
S26_OV(JHUZ)	Ovarian cancer CPTAC study S026 data generated at Johns Hopkins University (JHUZ)
S26_OV(PNNL)	Ovarian cancer CPTAC study S026 data generated at Pacific Northwest National Laboratory
S37_COAD(PNNL)	Colon Cancer CPTAC study S037 data generated at Pacific Northwest National Laboratory
S37_COAD(VU)	Colon Cancer CPTAC study S037 data generated at Vanderbilt

	University
S29_BRCA	CPTAC breast cancer study S29 – Discovery cohort
S39_BRCA	CPTAC breast cancer study S39 – Confirmatory cohort
S38_OV(JHUZ)	Ovarian cancer CPTAC study S038 data generated at Johns Hopkins University (JHUZ)
S38_OV(PNNL)	Ovarian cancer CPTAC study S038 data generated at Pacific Northwest National Laboratory(PNNL)
S43_UCEC	CPTAC Uterine Corpus Endometrial Carcinoma study S43
S44_CCRCC	CPTAC Clear Cell Renal Cell Carcinoma study S44

3. I was unable to find legends for the supplementary material.

2-3) Revision: Thanks for the good advice. We added legends for supplementary figures and tables in revised manuscript.

4. The data produced by this study – the tables of non-coding sequence–derived peptides – is an excellent resource! Mention that fact somewhere – it’s an important contribution of this work and should be addressed.

2-4) Revision: Thanks for your positive feedback. We agree with you it is an important resource for the community. We therefore emphasized this data resource in discussion in revised manuscript with “Importantly, we have shared all identified non-coding genes encoded peptides in healthy tissues and cancer samples, enabling others explore our data for further research.”

Introduction:

1. “It is conventionally believed...deleterious mutations” (line 40-41) – It would be best to cite a source here.

2-5) Revision: Thanks for the good advice. We added reference here [Proudfoot, N. Pseudogenes. Nature 286, 840–841 (1980)] in revised manuscript.

Results:

1. I would like to see a little bit more of a description of the sources of the healthy tissue – I had to go to the supplement of the original paper to see things like gender and age of

the patients. This could easily go in Table S1. Also, cite PRIDE database and CPTAC data portal here, and list the five cancer types – readers won't want to have to go to a supplementary table to see what types of cancer are being covered here. Consider replacing one of the main tables in the manuscript with a table summarizing the data used (at minimum, cancer types, normal tissue types, number of samples of each).

2-6) Revision: Thanks for the good advice. We made a new figure (figure 1a) to illustrate the information of samples used, including cancer types and number of samples of each. Normal tissue types were shown in figure 1c. We also supplemented more detailed clinical information in table S1 in revised manuscript.

2. Consider moving Figure S1 to the main manuscript and one of the tables (perhaps Table 1 – the data in Table 1 is highlighted in Table S2) into the supplement. The pipeline, while not the main focus of the paper, is something I imagine readers would like to be able to easily refer to.

2-7) Revision: Thanks for the good advice. we move Table 1 in Table S2 (create another sheet) to the supplement in revised manuscript. As for figure S1(workflow figure), we kept it in supplement as it is mostly the same to our previously published workflow, the changes have been described in Methods.

3. In Figure 1B, and other figures in the manuscript, samples could be labeled in a more easily readable way. For instance, “COAD_s37_PNNL” is not especially informative, especially since “PNNL” is not referred to anywhere else in the manuscript. (Maybe something like “COAD (CPTAC) Sample 37”?) Generally, it is best to avoid using abbreviations that are never defined.

2-8) Revision: Thanks for the good advice. “COAD_s37_PNNL” means Colon Cancer CPTAC study S037 data generated at Pacific Northwest National Laboratory. “PNNL”, “JHUZ” and “VU” mean different laboratories. They collaborated to produce this data. And we made an abbreviations table “Table of non-standard abbreviations” in revised manuscript.

See revision 2-2 for details.

4. “This result indicates...genomic regions” (line 90-91) – A significance measure would be good here.

2-9) Revision: Thanks for the good advice. We added a paired t test between normal and tumor, and there was no difference.

class	normal	tumor		normal	tumor
Downstream	2	3		0.003289	0.00123
Intronic-exonic boundary	2	5		0.003289	0.00205
Upstream	5	6		0.008224	0.00246
Retroelements	11	72		0.018092	0.02952
Intronic	17	69		0.027961	0.02829
Exonic	18	29		0.029605	0.01189
UTR region	24	31		0.039474	0.01271
lncRNA	103	180		0.169408	0.073801
Pseudogene	426	2044		0.700658	0.838048

```
> t.test(df$normal,df$tumor,paired = T)

Paired t-test

data: df$normal and df$tumor
t = -3.2054e-16, df = 8, p-value = 1
alternative hypothesis: true difference in means is not equal to 0
95 percent confidence interval:
 -0.04645253  0.04645253
sample estimates:
mean of the differences
 -6.457026e-18
```

We have updated the sentences in the manuscript.

" In t-test paired comparison, tumor and normal tissues showed no significant differences in percentage of expressed novel coding loci detected in different genomic regions."

5. “Our further analysis discovered...derived from pseudogenes” (line 92-94) – Please go into a little more detail. Can be as simple as the name of the software tool used to determine this.

2-10) Revision: We have now made more detailed annotations of the discovered novel coding loci. Peptides mapped to Intronic and Exonic specifically refers to those located in protein-coding genes' intronic and exonic region, and there are no annotated pseudogenes/lncRNAs at this locus. The ANNOVAR software tool was used to annotate novel peptides's genomic positions. Please note that if novel peptide belongs

to pseudogene, we will not count it again in other categories. Detailed annotation have added into figure 1d legend as following:

"Pseudogene: all categories of pseudogene (if novel peptide belongs to pseudogene, we will not count it again in other categories).

lncRNA: ncRNA

Exonic: coding gene's exon, not at canonical reading frame

Intronic: coding gene's intron

Intronic-exonic boundary: peptide spanning over coding gene's exon-intron boundary

UTR region: Untranslated region of coding gene

Upstream: Upstream of coding gene(1kb distance to closest UTR)

Downstream : Downstream of coding gene (1kb distance to closest UTR)

6. How were the samples shown in Figure S2A chosen?

2-11) Revision: If there are multiple samples for a tissue type, we combined the results together.

7. “Previous proteogenomic studies have rarely performed quantitative analysis” (line 103) – Please be more specific as to the type of proteogenomic studies referred to here (the world of proteogenomics is vast and there are tons of quantitative analyses out there), and perhaps cite one or two that have done quantitative analysis.

2-12) Revision: Thanks for the good advice. The quantitative analysis here refers to quantitative analysis of non-coding gene encoded peptides. Previous studies mostly report identification, lacking quantitation. We revised the main text that “Previous proteogenomic studies have mainly characterized protein level alterations from genomics aberrations including copy number variations and missense mutations (26–28). Our recent work investigated tissue specific expression of non-coding gene encoded peptides in five different human tissues(4). Here we extended the analysis in a comprehensive proteomics data of 31 different tissues(5). The identified novel peptides were quantified by MS1 maximum peak intensity using moFF(22).”

8. Suggestion for Figure 1E: separate the three groups (ubiquitous/nonspecific/tissue specific) somehow.

2-13) Revision: Thanks for the good advice. We separated the three groups (Figure 1E move to Figure 2) in revised manuscript. We marked ubiquitous, non-specific and tissue specific in figure 1e (now move to Figure 2).

Legend: Figure 2. Heatmap of MS1 intensity of non-coding gene encoded peptides detected in normal tissues.

9. It would be interesting to know the specific proportions of the three groups.

2-14) Revision: Thanks for the good advice. We added the specific numbers of the three groups with “Our analysis identified three groups of novel coding loci expression: 12 ubiquitous (expressed in at least 15 tissues), 93 non-specific (expressed in 2 to 15 tissues, robustly translated in one or two tissues but frequently translated at lower levels in other tissues.), and 114 tissue-specific expression (expressed in one tissue) (Figure 2). This pseudogene expression profile is different from the RNA-seq study, where the majority of expressed pseudogenes were identified as non-specific⁷. We speculated that many non-specific and lowly expressed pseudogenes was stochastically detected in tissues with only one sample analyzed (24 of 31 tissues have only one sample), consequently increased the number of tissue specific pseudogenes here (All data were included in supplementary table S2, with representative tissue-specific pseudogenes/lncRNAs highlighted).”

10. If you take the suggestion in #2 above, Table 1 can either be moved to its own sheet within Table S2, or you can point out that the representative pseudogenes/lncRNAs are highlighted in Table S2.

2-15) Revision: Thanks for the good advice. We moved table 1 as seperated sheet in table S2 and highlighted representative tissue-specific pseudogenes/ lncRNAs in revised manuscript.

11. “We detected two...eight unique peptides).” (line 112-116) – The reader can figure out that this means exclusively normal tissue, but it might be helpful just to mention that specifically.

2-16) Revision: Thanks for the good advice. We have clarified they were detected exclusively in normal tissue.

12. Suggestion for Figure 2: split into two panels – one for the bar graph and one for the matrices. This will allow for more specific references to the two parts of the figure.

2-17) Revision: Thanks for the good advice. We discard the old figure 2, because the bar plot is too complicated to interpret. it shows inter-datasets overlap, which is mostly related to instrumentation and methods. To compare the overlap of detected non-coding gene translation in different samples and datasets better. We made a new plot in figure 3 to illustrate it.

These sentences are that “we analyzed the overlap of detected novel coding loci in different samples within each study in figure 3a. The novel coding loci are divided into four groups by the percentage of samples they were identified. For example, the dataset PXD002619 produced the largest number of novel coding loci, but two third were identified in less than 25% samples. On average, one third of all novel loci were identified in more than 50% of samples.

Among different cancer datasets (in total 13 datasets covering five cancer types), 46%

of pseudogene identifications were repeatedly detected in at least two datasets. In comparison, only 16% of non-pseudogenes were detected in more than one dataset. Further analysis showed that 93% of pseudogenes identified commonly in 8 to 13 different datasets belong to housekeeping genes, which suggest pseudogenes derived from house-keeping genes are also ubiquitously expressed in different cancer types. (Figure 3b)” in revised manuscript.

Figure 3

a) the overlap of detected novel coding loci in different samples

b) overlap of pseudogenes between datasets

overlap of non-pseudogenes between datasets

Legend: Figure 3. a) the overlap of detected novel coding loci in different samples. For example, red bar indicates the number of novel coding loci that are only missing in 0%-25% of the samples, while blue bar indicates the number of novel coding loci missing in 75%-100% of the samples. b) overlap of pseudogenes and non-pseudogenes between datasets. 54% of pseudogenes only appear in one dataset, and 40.8%

pseudogenes appear in 2 to 7 data sets, and 5.2% pseudogenes appear in 8 to 13 data sets. While 97% of pseudogenes appearing in 8 to 13 data sets belong to housekeeping genes. 84.4% of non-pseudogenes only appear in one data set with higher specificity.

13. I would be more interested to see a version of Figure 2 with the ubiquitous pseudogenes removed – it makes sense that the ubiquitous pseudogenes would overlap, but what pseudogenes are specific to one or more cancer types as opposed to normal tissue? Along those lines, what happens if you remove the pseudogenes that are tissue-specific or non-specific to the normal tissue that gives rise to the particular cancer type (e.g. if you only look at BRCA pseudogenes that are not also normal breast-specific pseudogenes)?

2-18) Revision: Thanks for the good advice. We have now checked the overlap of non-housekeeping pseudogenes between different datasets (figure 3b). The overlap of non-housekeeping pseudogenes between datasets was much lower compared to these housekeeping pseudogenes.

14. For peptides that are “recurrently detected”, does this mean they were detected in multiple samples of each cancer type, or just in one or two samples of each cancer dataset? If the latter, I’m not sure it can be said that they’re cancer-specific rather than patient-specific.

2-19) Revision: Thanks for the good advice. We illustrated the sample proportion that “recurrently detected” *RHOXF1P3* and *MCTS2P* detected in different datasets (figure s3) with “Apart from the ubiquitously expressed pseudogenes, many pseudogenes were recurrently detected in specific cancers. The notable examples are *RHOXF1P3* (RhoX homeobox family member 1 pseudogene 3 and *MCTS2P* (malignant T cell amplified sequence 2 pseudogene) which are repeatedly detected from independent datasets of breast and ovarian cancers (Figure S3). The parental gene of *RHOXF1P3*, *RHOXF1*, is thought to inhibit cell apoptosis by activation of *BCL-2*(32). *MCTS2* is an imprinted gene and only paternally expressed retrogene copy (33)”. They are cancer-specific

rather than patient-specific.

In addition to above analysis, we made a new figure (figure 3) to compare the overlap of detected non-coding gene translation in different samples and datasets better.

See the responses to 2-17) for details.

15. Are the recurrently detected pseudogenes in Table 2 also detected in their analogous normal tissue? For instance, is *RHOXF1P3* just a pseudogene that is translated in breast and ovarian tissue, or is it specific to breast and ovarian cancer?

2-20) Revision: Thanks for the good advice. Pseudogene *RHOXF1P3* is NOT tissue specific pseudogene, it is detected in the 31 healthy tissues including breast, ovarian, pancreas, and testis tissue, but with higher expression in a subset of breast tumor samples.

16. The questions in 14 and 15 also apply to Table 3.

2-21) Revision: Our old statement about the lncRNAs in Table 3 is that they were identified multiple times in different cancer datasets, but not in the 31 healthy tissues. However, as the first review pointed out overlap between different studies can be cofounded by instrumentation and methods. Therefore we removed both Table 2 and Table 3, and described specific examples in the manuscript instead.

17. I don't think the claim in lines 152-153 that *DGCR5* probably encodes a functional protein product is supported by the data

2-22) Revision: Thanks for the good advice. It may need more evidences to support it. We revised the statement with "Our results present the first evidence that a potential novel coding locus in *DGCR9*'s last intron may encode a protein product in CCRCC". We have added a supplementary figure s3b to illustrate the genomic positions of detected novel coding locus at *DGCR9* intron. (Note: based on latest GENCODE32 annotation, this locus is updated to ncRNA *DGCR9*)

18. A small change in line 170 – I would change “probably” to “may”. The reason for this is that there is a distinct possibility that there actually is a protein product of ATP8A2P1, but it was unable to be detected by MS.

2-23) Revision: Thanks for the good advice. We revised the statement that “Another example is the breast cancer pseudogene ATP8A2P1 which showed high expression at RNA level^{7,8} was not detected at protein level in any of the breast cancer proteomics data, suggesting this pseudogene **may** only exert functions at RNA level. ”.

19. I strongly suggest not imputing missing values in the COAD peptide abundance data (line 179-180). Instead, leave the values missing, or remove those rows. This keeps the data as accurate as possible.

2-24) Revision: Thanks for the good advice. To keep the data as accurate as possible, we didn’t impute missing values in PXD002147 CRC (figure 4a) in revised manuscript. These sentences that described it “We investigated if certain pseudogenes/lncRNA encoded peptides have elevated expression in tumors in a colorectal cancer (CRC) dataset with 8 paired colorectal cancer samples and matched normal tissues (PXD002137)(35). In this data, 73 pseudogenes/lncRNA were identified supported by multiple peptides. An unsupervised clustering of 73 pseudogenes/lncRNA by the centered log₂ intensity is shown (Figure 4a).”

a) heatmap of colorectal cancer (PXD002137)

Legend: a) heatmap of colorectal cancer (PXD002137). Heatmap were scaled by row value.

20. The tab in Table S3 is mislabeled – should say it refers to Figure 3A.

2-25) Revision: Thanks for the good advice. We revised it.

21. A paired t-test for significance should be performed and reported for the data in Figure 3B. Also, using a box-and-whisker plot to present data with matched samples is a bit misleading, as it implies that the data points are independent – see Figure 2 in <https://doi.org/10.1371/journal.pbio.1002128> for ideas on presenting this in a more informative way.

2-26) Revision: Thanks for the good advice. We performed paired t-test on figure 4b in revised manuscript (old figure 3b moved to figure 4b). These sentences that described the method “Paired t-test analysis found 11 of the pseudogenes/lncRNAs were significantly upregulated in tumors compared to matched normal tissues. For example,

lnc-KMT5B-20:1, *lnc-NANOGP8-26:6* and *RP11-351N4.2* are up-regulated in colorectal cancer compared to matched normal tissues (Figure 4b).”

Legend: b) boxplot of non-coding genes significantly differentially expressed between tumor and normal (paired t-test, $p\text{-value} < 0.05$).

22. Again, please report significance for the data in Figures 3C-E.

2-27) Revision: Thanks for the good advice. We have now reported significance for the data in Figures 4C-E in revised manuscript (old figure 3 moved to figure 4).

23. “This contradicts...normal tissues” (line 189-190) I’m not sure this is true. I looked at the paper cited here and it appears that they also show increased expression of

DCGR5 RNA in cancer as compared to normal tissue.

2-28) Revision: Thanks for the good advice. We checked the cited paper and “LINC00037 was aberrantly up-regulated in ccRCC”. But this paper seems a little contradictory. Their title and results stated contradictory effect of LINC00037. We don't refer to it anymore. Therefore, we revised it with “the peptides encoded by lncRNA lnc-SERPIND1-41:10 (DGCR9 intron) showed significantly higher expression level in CCRCC compared to adjacent normal tissues (Figure 4c)”

24. I think the paragraph about RHOXF1P3 (line 198-209) needs more support. For instance, the confirmatory set has roughly the same proportion of RHOXF1P3-expressed samples in the normal and tumor sets. Also, what are the ER statuses of the samples in the ovarian and breast confirmatory sets?

2-29) Revision: Thanks for the good advice. We tried a fisher's exact test. The p-value is not significant enough. Indeed, more evidence is needed to prove that RHOXF1P3 pseudogene is associated with estrogen receptor signaling. Therefore, we modified our statement that “In the two CPTAC breast cancer datasets, the pseudogene RHOXF1P3 was identified with eight and seven unique peptides respectively, covering 89% of amino acid sequences of the open reading frame encoded by this pseudogene (Figure 5a). More interestingly, the pseudogene RHOXF1P3 encoded peptides were up-regulated (2 to 16-fold) in a subset of breast cancer patients both in CPTAC breast cancer Discovery and Confirmatory cohort (Figure 5b,5c) (27). Besides, RHOXF1P3 encoded peptides were also detected in two ovarian cancer patients (Figure 5d). We then analyzed the expression of RHOXF1P3 in a published RNA-seq data including 63 breast tumors and 10 adjacent normal tissues, which also showed up-regulated expression of RHOXF1P3 in tumor samples (Figure 5e). Together, our results demonstrated that pseudogene RHOXF1P3 is not only translated, but also upregulated in a subset of breast tumors.”

25. Significance value in Figure 4F?

2-30) Revision: Figure 4F was to show the detected novel peptides from 5' UTR of STARD10 and their expression level in different subtypes as heatmap. No significance test was used here.

26. Make sure to reference Figure 5 somewhere in the manuscript, and add significance values.

2-31) Revision: Thanks for the good advice. We added significance values for figure 6(old figure 5 moved to figure 6) in revised manuscript.

Legend: Figure 6. Relative abundance of LINE-1 ORF1 encoded peptides in tumor and normal. The heatmap shows the log₂ relative abundance of LINE-1 ORF1 encoded peptides in five different cancers. Grey boxes indicate missing values.

27. Line 256: should reference Table S5, not Table S6.

2-32) Revision: Thanks for the good advice. We corrected this error in revised manuscript.

28. For readers unfamiliar with parameters used in NetMHCpan and NetCTLpan, please explain what the thresholds mentioned in line 258 refer to.

2-33) Revision: Thanks for the good advice. Old manuscript description was not clear. The new version of NetCTLpan server uses a neural network to predict the binding of MHC peptides, including the function of the NetMHCpan server. Therefore, we

discarded the prediction of NetMHCpan server in revised manuscript. In the neoantigen prediction result and method section, we explained why we choose NetCTLpan tool. These sentences that “T cells recognize and bind to a peptide–MHC complex is a complex process with many crucial steps. First, the abnormal proteins are hydrolyzed by proteases into peptide fragments in the cytoplasm, and then peptide fragments transported by the transporter associated with antigen processing (TAP) protein into the endoplasmic reticulum, where the peptide bind to an MHC molecule³⁷. The NetCTLpan server integrates predictions of proteasomal C terminal cleavage, TAP transport efficiency, and MHC class I binding affinities, which takes into account the antigen processing and presentation³⁸. Therefore, we used NetCTLpan server to predict the neoantigen.” were added into revised manuscript.

In addition, detailed thresholds selection was explained as following:

“The predicted values were calculated as the weighted average of the MHC, TAP and C-terminal cleavage scores, and as %-Rank to a set of 1000 random natural 9mer peptides. The prediction parameters of NetCTLpan are set as follows: the sorting threshold $\leq 0.5\%$ (%Rank of prediction score to a set of 1000 random natural 9mer peptides), the weight proportion of c-terminal amino acid residues was 0.225, and the weight proportion of TAP transport efficiency was 0.025.”

29. Figure 6B is difficult to interpret, and may be better off either replaced with a simple Venn diagram (overlap between MHC and CTL) or moved to the supplement (or both).

2-34) Revision: Thanks for the good advice. Old figure 6B is difficult to interpret. Therefore, we revised this part of predicting neoantigens. New version of NetCTLpan server uses a neural network to predict the binding of MHC peptides, including the function of the NetMHCpan server. Therefore, we discarded the prediction of NetMHCpan server in revised manuscript. We discarded the old figure 6, and kept the prediction result in a supplementary table S5.

30. Table S5, as cited in line 261, doesn't really lay out the overlap – maybe add a third sheet to that Excel file that contains only the overlap novel loci?

2-35) Revision: Thanks for the good advice. As in the previous response, we discarded the prediction of NetMHCpan server in revised manuscript. So table S5 just have the result of NetCTLpan predicted.

31. How were the antigen peptides shown in Figure 6C chosen?

2-36) Revision: In old figure 6C, the antigen peptides were chosen from old figure 6b, which were bind with three HLA allele in both software. They have higher credibility. But please note that we deleted old figure 6 and changed the criteria of neoantigen prediction.

Discussion:

1. “According to...non-coding regions” (line 294-296) – I’d suggest moving this to the next paragraph.

2-37) Revision: Thanks for the good advice. Suggestions have been taken and we added a new paragraph to discuss tumor specific neoantigen.

We modified and expanded these sentences that “According to the predicted results by the NetMHCpan4.0 and NetCTLpan, there are a large number of potential tumor specific antigens from the non-coding regions.” as “Mutation derived tumor specific neoantigen could be recognized by infiltrated T cells and trigger its cytotoxic activity against tumor cells. The neoantigen based immunotherapy has been tested and shown promising results in clinical trials⁴⁶. However, missense mutation derived neoantigens may have limited immunogenicity due to its non-self feature relies on only one amino acid difference. In comparison, neoantigens derived from non-coding gene encoded peptides that possess larger sequence differences hold greater promises due to its high potential immunogenicity⁴⁷. **According to our analysis by the NetCTLpan, there are a large number of potential tumor neoantigens predicted from the non-coding genes encoded peptides.** In the aspect of clinical applications, these peptides encoded by non-coding regions, especially the ones specific to tumors, may provide a new class of tumor neoantigens possessing more potent immunogenic activity to be explored as cancer vaccine 10.”

2. “Our results suggest...normal and tumors” (line 298-299) – Please elaborate on how the results suggest this.

2-38) Revision: Due to lack of direct data support, we removed this statement.

Reviewer #3 (Remarks to the Author):

The authors studied published proteomics data from 926 cancer samples of five cancer types and 31 different healthy human tissues to describe the translation of pseudogenes. Although I find this work relevant and contributes to the still very unknown field of pseudogenes translation I can not recommend it for publication until the authors work on the following aspects:

- What is the statistical criteria to choose 31 healthy human tissues? Can the authors describe further the criteria the used for the selection of those matched 31 tissues?

3-1) Revision: The 31 healthy human tissues were used to search non-coding gene encoded peptides in diverse tissue types. It is so far one of most comprehensive human tissue proteomics datasets that has been published and made accessible.

- The authors suggest that there is a common mechanism that regulates the translation of pseudogenes in healthy and tumor tissues, especially those ones involved in cytoskeleton. How can the authors link this possible mechanism to the upregulation of many cytoskeleton proteins in several cancer types?

3-2) Revision: Due to lack of direct data support, we removed this statement.

- The authors describe throughout the paper examples of pseudogenes in specific cancers, *RHOXF1P3*, *PA2G4P4*, *MCTS2P*, *DGCR5*, and so on. However there is a very poor description of these genes and their products and no discussion at all how they might function in those specific cancers.

3-3) Revision: we added the description of these genes. These sentences are "The notable examples are *RHOXF1P3* (RhoX homeobox family member 1 pseudogene 3 and *MCTS2P* (malignant T cell amplified sequence 2 pseudogene) which are repeatedly detected from independent datasets of breast and ovarian cancers (Figure S3). The parental gene of *RHOXF1P3*, *RHOXF1*, is thought to inhibit cell apoptosis by activation of BCL-2(32). *MCTS2* is an imprinted gene and only paternally expressed retrogene copy(33)."

- What is the rationale of using the left-shifted Gaussian distribution method to carry out imputations? What is the percentage of missing values? Did the authors try other imputation methods to address low abundance?

3-4) Revision: We explored the pattern of missing values. The peptides with missing values have on average low intensities. Therefore, we used the left-shifted Gaussian distribution method. However, to keep the data as accurate as possible, we didn't impute missing values in PXD00217 CRC (figure 4a) in revised manuscript.

These sentences that described it “We investigated if certain pseudogenes/lncRNA encoded peptides have elevated expression in tumors in a colorectal cancer (CRC) dataset with 8 paired colorectal cancer samples and matched normal tissues (PXD002137)(35). In this data, 73 pseudogenes/lncRNA were identified supported by multiple peptides. An unsupervised clustering of 73 pseudogenes/lncRNA by the centered log2 intensity is shown (Figure 4a).”

a) heatmap of colorectal cancer (PXD002137)

Legend: a) heatmap of colorectal cancer (PXD002137). Heatmap were scaled by row value.

- Can the authors support the findings of RHOXF1P3 expression in ER positive tumors with statistical evidence on the main text?

3-5) Revision: Thanks for the good advice. We tried a fisher's exact test. The p-value is not significant enough. Indeed, more evidence is needed to prove that RHOXF1P3 pseudogene is associated with ER positive tumors. We modified our statement that

“In the two CPTAC breast cancer datasets, the pseudogene RHOXF1P3 was identified with eight and seven unique peptides respectively, covering 89% of amino acid sequences of the open reading frame encoded by this pseudogene (Figure 5a). More interestingly, the pseudogene RHOXF1P3 encoded peptides were up-regulated (2 to 16-fold) in a subset of breast cancer patients both in CPTAC breast cancer Discovery and Confirmatory cohort (Figure 5b,5c) (27). Besides, RHOXF1P3 encoded peptides were also detected in two ovarian cancer patients (Figure 5d). We then analyzed the expression of RHOXF1P3 in a published RNA-seq data including 63 breast tumors and 10 adjacent normal tissues, which also showed up-regulated expression of RHOXF1P3 in tumor samples (Figure 5e). Together, our results demonstrated that pseudogene RHOXF1P3 is not only translated, but also upregulated in a subset of breast tumors.”

-The authors do not describe any difficulty/challenge/adjustments in the performance of the proteogenomics workflow they used. What is that the case?

3-6) Revision:

The following adjustments were made in the new workflow.

1. database indexing. Previously, database indexing is performed separately for each spectra input file. Database indexing is done only once in the new workflow, and all parallel processes share the same indexed database at the search step. It reduces a lot intermediate files and increases speed.
2. hg38 genomic coordinates. Previously, all peptides were mapped to hg19 genome assembly, now peptides were mapped to hg38 assembly.

Minor aspects:

- space after Z. on line 12

3-7) Revision: Thanks for the good advice. We revised it.

- The current format of supplementary tables does not help to retrieve information intuitively. For example, What is the meaning of the coloring used in the middle to supplementary table S2?

3-8) Revision: Thanks for the good advice. We added it in revised manuscript with “All data were included in supplementary table S2 with detailed annotation, representative tissue-specific pseudogenes/ lncRNAs color highlighted”

- Line 56, write the reference properly

3-9) Revision: These sentences that "Secondly, a published study by Laumont *et al.* detected more tumor specific antigens from non-coding regions compared to mutations in protein-coding regions" in first manuscript were changed into " Secondly, a published study by Laumont *et al.* detected more tumor specific antigens from non-coding regions compared to mutations in protein-coding regions(10). "

- Missing space before 3 on line 249

3-10) Revision: Thanks for the good advice. We revised it.

- Can the authors describe in detail the quantification method of the proteomic datasets?

3-11) Revision: We have added the description of quantification method in the Methods:

For label free quantification, the identified novel peptides were quantified by extracting MS1 maximum peak intensity using moFF, and the log₂ MS1 intensity of peptides belonging to a particular novel coding locus were summed. For other datasets based on TMT relative quantification, peptide log₂ ratio (using internal reference as denominator) is summarized into novel loci log₂ ratio by taking the median value of all peptides.

- Can the authors describe more in detail the NetCTLpan parameters and explain why they choose those parameters?

3-12) Revision: Thanks for the good advice. We explained why we choose NetCTLpan

software tool and described more in detail the NetCTLpan parameters as following:

“T cells recognize and bind to a peptide–MHC complex is a complex process with many crucial steps. First, the abnormal proteins are hydrolyzed by proteases into peptide fragments in the cytoplasm, and then peptide fragments transported by the transporter associated with antigen processing (TAP) protein into the endoplasmic reticulum, where the peptide bind to an MHC molecule(44). The NetCTLpan server integrates predictions of proteasomal C terminal cleavage, TAP transport efficiency, and MHC class I binding affinities, which takes into account the antigen processing and presentation(24). Therefore,we used NetCTLpan server to predict the neoantigen.”

“The prediction parameters of NetCTLpan are set as follows: the sorting threshold $\leq 0.5\%$ (%Rank of prediction score to a set of 1000 random natural 9mer peptides), the weight proportion of c-terminal amino acid residues was 0.225, and the weight proportion of TAP transport efficiency was 0.025. The predicted value is calculated as the weighted average of the MHC, TAP and C-terminal cleavage scores, and as %-Rank to a set of 1000 random natural 9mer peptides.”

- Poor format of the tables in the main text

3-13) Revision: We removed two tables in supplementary files and updated the table format.

- Correct know peptides by known peptides in Figure S1

3-14) Revision: Thanks for the good advice. We corrected it

Reviewers' comments:

Reviewer #1 (Remarks to the Author):

The authors have addressed my comments adequately. Thank you, this is an interesting paper and it has significantly improved.

Reviewer #2 (Remarks to the Author):

This manuscript details the use of a proteogenomics analysis workflow to find and analyze peptides encoded by non-coding sequences in publicly available datasets comprising five different cancer types and 31 healthy human tissues. The authors describe the protein expression of several pseudogenes and lncRNAs, including their tumor specificity (or lack thereof), and give several examples of how their data expands on previous studies of particular pseudogenes and lncRNAs. They also predict a quantity of potential neoantigens translated from tumor-specific noncoding regions that could be used in the future as bases for cancer vaccines. The questions addressed in the manuscript are fascinating and the authors present a wealth of good data and analysis. However, I have several concerns about the methods used; it is not always clear whether the correct subsets of the CPTAC dataset were used, and the authors do not appear to have manually validated any of their spectra, which is necessary when dealing with novel peptide detection. Additionally, I would recommend removing a section from the manuscript that does not have enough supporting data and does not seem relevant to the authors' stated goal.

Major points are denoted by ***

OVERALL

1. A copy editor would help improve the general readability and clarity of the paper.
2. ***CPTAC often has multiple centers run mass spec analysis on the same samples – for instance, the S038 OV JHU study is done on almost all of (if not all of) the same tumors as the PNNL study. The authors do not address this – how was it handled in the analysis? It is a concern when peptides are found in one version but not the other.
3. ***Did you look at any of the spectra from the spectral matches? Identifying novel peptides with mass spectrometry will invariably lead to a lot of false positives – for any spectra you think may be of interest, please manually confirm the annotations. Including the annotated spectra in the supplemental material would be ideal.

INTRODUCTION

4. Line 46: "It remains unknown if pseudogenes are translated in tumor tissues" The previous sentence contradicts this statement.

RESULTS

5. Line 63: The normal data from CPTAC isn't mentioned here, so it's a bit confusing to see it in Figure 1A.
6. Line 64: Cite PRIDE and CPTAC here.
7. Line 63-67: The addition of the table of abbreviations is welcome, but the five cancer types should be mentioned here as well – there is nowhere in the main paper that the authors actually list the

cancer types they are analyzing.

8. Line 73: The title in Reference #15 is incorrect (should not say "Figure 1")

9. Figure S1: Now that Figure 1A is in the main document, it is unnecessary to repeat that bar graph here.

10. ***Line 80: It is unclear here and at several other points in the paper whether the entire CPTAC dataset is used when referring to the "cancer dataset" (including normal, QC, reference, and other non-tumor samples), or if just the tumor samples were used. Please ensure that the tumor samples were the only samples used.

11. Figure 1C: Separate the Venn diagram into its own panel.

12. Figure 1D: Change the y-axis labels to a log scale.

13. Line 96: What is being paired in this paired t-test?

14. Figure 1E: "Tubulin" is misspelled (also in Figure S2A)

15. Figure 1E: The figure should include a category for "Other" if these eight categories do not comprise the entire set of novel peptides, otherwise the percentages are inaccurate.

16. ***Line 115-159/Figure 2: This entire section does not fit well in the paper – if the main focus of the paper is the expression of non-coding genes in the cancer genome, there doesn't need to be a section about the normal tissue dataset that is not related to the cancer analysis. Consider removing it altogether.

17. ***Line 129-147: It is very difficult to say anything definitive about a tissue type for which only one sample has been analyzed. The peptides found in the fallopian tubes, salivary gland, and testis are just as likely to be specific to the individual as they are to be specific to the tissue type. (The fact that GTex backs up the testis peptides is good, but it also means this is not a novel finding.)

18. Line 144: Were these found in both pituitary tissue samples, or only in one? If just one, the same concern as above applies.

19. ***Line 167-169: Because the studies are done largely in duplicate (see point #2), any of the detections that were detected only in both versions of the S027, S037, or S038 datasets should be checked to make sure they were not just detected in samples that are repeated between the datasets.

20. Figure 3B: Better labeling needed on the central Venn diagram.

21. Line 177: The two ovarian datasets mentioned here are not independent – they consist of the same set of tumors.

22. Figure S3A: Separate RHOXF1P3 and MCTS2P into separate bar graphs. It is confusing to have them together.

23. Figure S3A: Should be in the main figures, not the supplement.

24. Line 179-180: It is a concern that MCTS2P only appears in one of the two duplicate datasets. Manually validating the spectra in which its peptides appear would show that this is due to sample processing on the part of JHU/PNNL rather than a false positive in the spectral matching.

25. Line 182-184: How many samples were these ten peptides found in? If they were all found in one sample, it would be more likely to be an individual variation; if they were found in several samples, then this finding would be quite interesting.

26. ***Figure 4C: QC/NCI7 samples are for housekeeping only. They should not be included in any analysis.

27. Line 217-225: The p-values here should have multiple testing corrections. (Most should still be significant, but it is best to be rigorous.)

28. Figure 5A: It appears that the first highlighted amino acid (6R) should not be highlighted.

29. Figure 5D: Remove QC samples from the analysis.

30. Line 255-257: According to Figure 6, it is not true that the LINE-1 ORF1p encoded peptides were always significantly higher in tumors compared to normal samples – this was only true of three

tumor types.

31. Figure 6: Remove QC samples from the analysis.

32. Table 1: How were the peptides chosen for the main manuscript table? As only two peptides are actually discussed here, Table 1 can likely be removed.

METHODS

33. Line 335: Mention the matched normal tissue here.

Reviewer #3 (Remarks to the Author):

Dear authors,

I am satisfied with the reply to my questions and concerns and I can finally understand and read your tables.

Dear reviewer,

Thank you very much for the advice on correcting my manuscript entitled **“Quantitative proteogenomics detects increased expression of non-coding genes encoded peptides in cancer genome”**. And we revised the manuscript thoroughly based on the comments. In the revised manuscript, all the suggestions and comments have been taken into consideration carefully. Here are point-to-point answers to the questions.

Reviewer #2 (Remarks to the Author):

This manuscript details the use of a proteogenomics analysis workflow to find and analyze peptides encoded by non-coding sequences in publicly available datasets comprising five different cancer types and 31 healthy human tissues. The authors describe the protein expression of several pseudogenes and lncRNAs, including their tumor specificity (or lack thereof), and give several examples of how their data expands on previous studies of particular pseudogenes and lncRNAs. They also predict a quantity of potential neoantigens translated from tumor-specific noncoding regions that could be used in the future as bases for cancer vaccines. The questions addressed in the manuscript are fascinating and the authors present a wealth of good data and analysis. However, I have several concerns about the methods used; it is not always clear whether the correct subsets of the CPTAC dataset were used, and the authors do not appear to have manually validated any of their spectra, which is necessary when dealing with novel peptide detection. Additionally, I would recommend removing a section from the manuscript that does not have enough supporting data and does not seem relevant to the authors' stated goal.

Major points are denoted by ***

OVERALL

1. A copy editor would help improve the general readability and clarity of the paper.
Response: Thanks for the good advice. We have made a few changes in the manuscript to improve the language.
2. ***CPTAC often has multiple centers run mass spec analysis on the same samples – for instance, the S038 OV JHU study is done on almost all of (if not all of) the same tumors as the PNNL study. The authors do not address this – how was it handled in the analysis? It is a concern when peptides are found in one version but not the other.

Response: Yes, some samples indeed were analyzed at multiple centers in CPTAC data sets. Among the data we quoted, there are three datasets analyzed by two research centers, as shown in the table below. For example, the 174 ovarian tumors of S026 OV study, JHU and PNNL analyzed 122 and 84 tumors, respectively, and 32 tumors are common. The samples didn't completely overlap, and they were analyzed separately (shown separately in Figure 3).

Datasets	Type	Total	JHU	PNNL	Common
S026 OV	tumor	174	122	84	32
S037 COAD	tumor	104	100	97	93
	normal	100		100	
S038 OV	tumor	100	85	84	69
	normal	25	22	19	16

3. ***Did you look at any of the spectra from the spectral matches? Identifying novel peptides with mass spectrometry will invariably lead to a lot of false positives – for any spectra you think may be of interest, please manually confirm the annotations. Including the annotated spectra in the supplemental material would be ideal.

Response: We have plotted annotated spectra of peptides of interest (in total 123 spectra of 87 unique peptides) and provided as supplemental material. We looked through these annotated spectra, most of them have complete coverage of b, y ion series along the peptide chain, except for several long peptides.

INTRODUCTION

4. Line 46: “It remains unknown if pseudogenes are translated in tumor tissues” The previous sentence contradicts this statement.

Response: The previous sentence mentions that pseudogene translation has been detected in cancer cell line, while it has not been systematically investigated in different tumor types. We revised this sentence.

“However, it has not been systematically investigated if non-coding genes encoded peptides can be found in tumor tissues and whether translation of non-coding genes is a sporadic event or under certain specific regulation in different type of tumors.”

RESULTS

5. Line 63: The normal data from CPTAC isn't mentioned here, so it's a bit confusing to see it in Figure 1A.

Response: Thanks for the good advice. These sentences that “We downloaded proteomics data of 40 normal samples from 31 healthy tissues and 933 cancer samples

of five types from the PRIDE database and CPTAC Data Portal.” were changed into “We downloaded proteomics data of 40 normal samples from 31 healthy tissues, 933 tumor samples and 275 tumor adjacent normal samples from the PRIDE database and CPTAC Data Portal.”

6. Line 64: Cite PRIDE and CPTAC here.

Response: Thanks for the good advice. We have now cited PRIDE and CPTAC in line 64.

7. Line 63-67: The addition of the table of abbreviations is welcome, but the five cancer types should be mentioned here as well – there is nowhere in the main paper that the authors actually list the cancer types they are analyzing.

Response: Thanks for the good advice. These sentences “The five types of cancer are Breast Cancer (BRCA), Clear Cell Renal Cell Carcinoma (CCRCC), Colon Cancer (COAD), Ovarian cancer (OV) and Uterine Corpus Endometrial Carcinoma (UCEC)” were added into the manuscript.

8. Line 73: The title in Reference #15 is incorrect (should not say “Figure 1”)

Response: Thanks for the good advice. We have corrected it in the revised manuscript.

9. Figure S1: Now that Figure 1A is in the main document, it is unnecessary to repeat that bar graph here.

Response: Thanks for the good advice. We have removed the bar graph in revised figure s1.

10. ***Line 80: It is unclear here and at several other points in the paper whether the entire CPTAC dataset is used when referring to the “cancer dataset” (including normal, QC, reference, and other non-tumor samples), or if just the tumor samples were used. Please ensure that the tumor samples were the only samples used.

Response: Thanks for the good advice. It is indeed unclear. It is actually the entire CPTAC datasets.

For datasets based on label free absolute quantification (PXD002136 and S037 COAD(VU)), we can distinguish whether a novel peptide comes from a cancer or a normal sample. However, for datasets based on iTRAQ or TMT relative quantification (S29 BRCA, S038 OV (PNNL), S038 OV (JHU), S039 BRCA, S043 UCEC and S044 CCRCC), tumors, matched normal tissues QC, reference and other non-tumor samples were often mixed in one iTRAQ or TMT set. Currently, there is no method to conclude the source of the peptides in iTRAQ or TMT experiments. In the end, we decided to summarize the number of novel peptides without distinguishing the sample source.

We have made it clear in the manuscript and in figure 1c.

“The number of unique peptides per novel coding locus were summarized for 31 healthy tissues, and the CPTAC datasets (including both 933 tumor samples and 275

tumor adjacent normal samples) respectively (Figure 1b and 1c).”

11. Figure 1C: Separate the Venn diagram into its own panel.

Response: Thanks for the good advice. We have separated the Venn diagram into its own panel in revised figure 1d.

12. Figure 1D: Change the y-axis labels to a log scale.

Response: y-axis is actually in log₂ scale, but values on the bar were not log transformed.

13. Line 96: What is being paired in this paired t-test?

Response: The percentages of identified novel peptides in one genomic region (old figure 1d, new figure 1e) in the healthy tissues and the CPTAC datasets are being paired in this paired t-test. We have clarified this in corresponding place.

These sentences were that “In t-test paired comparison, the CPTAC datasets and the healthy tissues showed no significant differences in the percentage of novel coding loci detected in different genomic regions.”

14. Figure 1E: “Tubulin” is misspelled (also in Figure S2A)

Response: Thanks for the good advice. We have corrected it in revised manuscript.

15. Figure 1E: The figure should include a category for “Other” if these eight categories do not comprise the entire set of novel peptides, otherwise the percentages are inaccurate.

Response: Thanks for the good advice. We have added a new category for “Other” in figure 1f (replace figure 1e) as below. Please note we just stated the pseudogene category. And these sentences “The above eight major categories include 426 and 2044 novel peptides, comprising 70% and 84% total novel peptides detected from the healthy tissues and cancer datasets.” were changed into “The number of above pseudogene peptide include 428 and 1970 novel peptides, comprising 70.9% and 84.9% total novel peptides detected from the healthy tissues and cancer datasets.”

16. ***Line 115-159/Figure 2: This entire section does not fit well in the paper – if the main focus of the paper is the expression of non-coding genes in the cancer genome, there doesn't need to be a section about the normal tissue dataset that is not related to the cancer analysis. Consider removing it altogether.

Response: Thanks for the good advice. Although the healthy tissues were not our main focus of the paper, presenting such analysis could well indicate that translation of non-coding genes are not cancer specific events, many non-coding genes especially pseudogenes are ubiquitously expressed in different tissues.

17. ***Line 129-147: It is very difficult to say anything definitive about a tissue type for which only one sample has been analyzed. The peptides found in the fallopian tubes, salivary gland, and testis are just as likely to be specific to the individual as they are to be specific to the tissue type. (The fact that GTex backs up the testis peptides is good, but it also means this is not a novel finding.)

Response: Thanks for the good advice. We added a description of the relevant limitations in the Result and Discussion sections.

In Result: “We speculated that many non-specific and lowly expressed pseudogenes was stochastically detected in tissues with only one sample analyzed (24 of 31 tissues have only one sample), consequently increasing the number of tissue specific pseudogenes here”

In Discussion: “We acknowledge the limitation that it is not a complete representation of all tissue types and most of tissue types have only one sample. Thus, some tissue specific pseudogenes detected here may arise stochastically.”

18. Line 144: Were these found in both pituitary tissue samples, or only in one? If just one, the same concern as above applies.

Response: Pituitary tissue has two samples. GH1 and POMC non-canonical reading frame encoded peptides were found in both pituitary tissue samples. However, the annotated spectra of peptides from POMC indicate this is likely a false match. In the manuscript, we have just stated GH1 was detected in both pituitary tissue samples.

“Interestingly, in both pituitary tissue samples, peptides were identified from a lncRNA that overlap with the coding region of pituitary specific protein coding gene, GH1, but in non-canonical reading frame (see annotated spectra in supplementary material). Our data indicates the GH1 may have dual coding frames that encode unknown new proteins.”

Sequence	Gene Name	MS1 intensity of sample 1	MS1 intensity of sample 2
AAPHLPAAHPVVAGAR	GH1	414576373.7	104470272
GTEVFIPAEPDLPLFLR	GH1	2840347.844	6670478.594
SLYPKGTEVFIPAEPDLPLFLR	GH1	0	299553.0938
AASDREPPEVR	POMC	15212802	29044984.75
HGPLPLGPIRPPQQQQR	POMC	0	1480616.25

19. ***Line 167-169: Because the studies are done largely in duplicate (see point #2), any of the detections that were detected only in both versions of the S027, S037, or S038 datasets should be checked to make sure they were not just detected in samples that are repeated between the datasets.

Response: Thanks for the good advice. This figure is used to show many of the peptide encoding pseudogenes were not solely identified in one dataset, in comparison most of non-pseudogenes were identified in one dataset, despite several datasets actually share many common samples.

20. Figure 3B: Better labeling needed on the central Venn diagram.

Response: Thanks for the good advice. We have made the labelling clearer.

21. Line 177: The two ovarian datasets mentioned here are not independent – they consist of the same set of tumors.

Response: Thanks for the good advice. The 100 ovarian tumors of S038 OV study,

JHU and PNNL analyzed 85 and 84 tumors, respectively, and 69 tumors are common.

22. Figure S3A: Separate RHOXF1P3 and MCTS2P into separate bar graphs. It is confusing to have them together.

Response: Thanks for the good advice. We have separated RHOXF1P3 and MCTS2P into separate bar graphs in figure 3c.

Figure 3c legend: “c) The percentage of samples detected with RHOXF1P3 and MCTS2P.”

23. Figure S3A: Should be in the main figures, not the supplement.

Response: Thanks for the good advice. We have moved figure S3A to the main figure 3c.

24. Line 179-180: It is a concern that MCTS2P only appears in one of the two duplicate datasets. Manually validating the spectra in which its peptides appear would show that this is due to sample processing on the part of JHU/PNNL rather than a false positive in the spectral matching.

Response: Thanks for the good advice. We have manually validated the spectra of MCTS2P. From the annotated spectra, combining matched b ions and y ions together could support the identified amino acid sequence of two peptides from MCTS2P in S38_PNNL dataset.

25. Line 182-184: How many samples were these ten peptides found in? If they were all found in one sample, it would be more likely to be an individual variation; if they were found in several samples, then this finding would be quite interesting.

Response: The ten peptides are detected in different samples. Figure 4c showed the identified peptides of DGCR9 in different samples. DGCR9 encoded peptides were found in most CCRCC samples.

26. ***Figure 4C: QC/NCI7 samples are for housekeeping only. They should not be included in any analysis.

Response: Thanks for the good advice. We have corrected it in revised figure 5a.

27. Line 217-225: The p-values here should have multiple testing corrections. (Most should still be significant, but it is best to be rigorous.)

Response: Thanks for the good advice. The p-values have been multiple testing corrected using “BH” correction. The adjusted values are updated.

28. Figure 5A: It appears that the first highlighted amino acid (6R) should not be highlighted.

Response: Thanks for the good advice. We have corrected it in revised figure 5a.

a) >RHOXF1P3 | ENST00000640298.1 Frame 0 from nucleotide 653 to 1049 (132 aa)
MGSQPRSGHEDTGNPGLGLFHEHQEGDNAKAVGTLVTEAGGDEEKKIRSKPEQGAGAE 60
ESHFCAGAADPTIKDNQKRGGGQEPSQQQPEASSPGLLRGGATTPILAASLQPQCQQV 120
APEGARQHFPTL 132

amino acids in red were supported by peptide evidence

29. Figure 5D: Remove QC samples from the analysis.

Response: Thanks for the good advice. We have corrected it in revised figure 5d.

30. Line 255-257: According to Figure 6, it is not true that the LINE-1 ORF1p encoded peptides were always significantly higher in tumors compared to normal samples – this was only true of three tumor types.

Response: Thanks for the good advice. We have corrected our description and these sentences are “The quantitative analysis showed higher expression of the LINE-1 ORF1p encoded peptides in UCEC, OV and COAD compared to their respective normal samples”

31. Figure 6: Remove QC samples from the analysis.

Response: Thanks for the good advice. We have corrected it in revised figure 6.

32. Table 1: How were the peptides chosen for the main manuscript table? As only two peptides are actually discussed here, Table 1 can likely be removed.

Response: Thanks for the good advice. Since Table 1 is part of supplementary table S6. We have removed it in the revised manuscript.

METHODS

33. Line 335: Mention the matched normal tissue here.

Response: Thanks for the good advice. We have added it. These sentences are “LC-MS/MS raw files of 40 normal samples from 31 healthy tissues, 933 cancer samples of five cancer types and 275 tumor adjacent normal samples were obtained from National Cancer Institute Clinical Proteomic Tumor Analysis Consortium (CPTAC) and PRoteomics IDentifications (PRIDE) database”

REVIEWERS' COMMENTS:

Reviewer #2 (Remarks to the Author):

The authors have satisfactorily addressed most of my comments. Overall I think the manuscript is much improved! While I continue to believe that the inclusion of “Proteomics detects ubiquitous and tissue-specific translation of pseudogenes and lncRNA” does not strengthen the paper, I ultimately leave the decision to the authors and the editors.

I thank the authors for including their annotated spectra in the most recent version. Upon looking at these spectra, however, I do have some concerns. While it is certainly important to ensure that most of the b and y ions are annotated when validating a spectrum, it is also important to ensure that most of the highest-intensity peaks are explained by the annotation. For instance, the authors state that three peptides were found to support GH1 in pituitary samples. However, two of the three peptides, AAPHLPAHPVAGAR and SLYPKGTEVFIPAEPDLPLFLR, have an abundance of unexplained tall peaks in their spectra, indicating that they are likely false positives (GTEVFIPAEPDLPLFLR looks a bit better). Annotating additional peaks such as neutral losses, c/z ions, and the +1/+2s may make manual validation of the spectra a little easier.

Dear reviewer,

Thank you very much for the advice on correcting my manuscript entitled **“Quantitative proteogenomics detects increased expression of non-coding genes encoded peptides in cancer genome”**. And we revised the manuscript thoroughly based on the comments. In the revised manuscript, all the suggestions and comments have been taken into consideration carefully. Here are point-to-point answers to the questions.

Reviewer #2 (Remarks to the Author):

The authors have satisfactorily addressed most of my comments. Overall I think the manuscript is much improved! While I continue to believe that the inclusion of “Proteomics detects ubiquitous and tissue-specific translation of pseudogenes and lncRNA” does not strengthen the paper, I ultimately leave the decision to the authors and the editors.

I thank the authors for including their annotated spectra in the most recent version. Upon looking at these spectra, however, I do have some concerns. While it is certainly important to ensure that most of the b and y ions are annotated when validating a spectrum, it is also important to ensure that most of the highest-intensity peaks are explained by the annotation. For instance, the authors state that three peptides were found to support GH1 in pituitary samples. However, two of the three peptides, AAPHLPAAHPVVAGAR and SLYPKGTEVFIPAEPDLPLFLR, have an abundance of unexplained tall peaks in their spectra, indicating that they are likely false positives (GTEVFIPAEPDLPLFLR looks a bit better). Annotating additional peaks such as neutral losses, c/z ions, and the +1/+2s may make manual validation of the spectra a little easier.

Response: Thanks to reviewer for the valuable suggestions that have greatly improved our manuscript.

We prefer to keep the section. Healthy tissues detect ubiquitous and tissue-specific translation of pseudogenes /lncRNA indicate that translation of non-coding genes are not cancer specific events, and many non-coding genes especially pseudogenes are ubiquitously expressed in different tissues.

We must admit the possibility of false-positives in our proteogenomics analysis,

although we have made some efforts to reduce the possibility of false-positives, including control the FDR of peptides according to the target-decoy strategy, remove the novel coding loci supported by only one peptide and perform blastp analysis to ensure that novel peptides do not match to the known reference protein database. Therefore, we added discussion about the possibility of false-positives in our manuscript. And we annotated additional peaks such as neutral losses, c/z ions in the supplementary figure annotated spectra.

“To avoid potential false positives, we only analyzed novel coding loci supported by at least two peptides, resulting majority of identifications removed (Figure 1). The inter-patient overlap of identified novel loci within each study showed approximately one third of them were detected in over 50% samples (Figure 3). Since false positives could be still present, we thus annotated MS2 spectra of 87 peptides from major findings described in the manuscript (Supplementary Data 3). The annotated spectra revealed several suspicious identifications, but most of them have well matched fragment ions.”